# Tumor microenvironment remodeling plus immunotherapy could be used in mesenchymal-like tumor with high tumor residual and drug resistant rate

Shuai Shen [1,4], Xing Liu[1,4], Qing Guo[1,4], Qingyu Liang[2], Jianqi Wu[1], Gefei Guan[2], Cunyi Zou[2], Chen Zhu[2], Zihao Yan[2], Tianqi Liu[1], Ling Chen[3], Peng Cheng [2✉], Wen Cheng [1✉] & Anhua Wu [1✉]

Epithelial-mesenchymal transition (EMT) is a common process during tumor progression and is always related to residual tumor, drug resistance and immune suppression. However, considering the heterogeneity in EMT process, there is still a need to establish robust EMT classification system with reasonable molecular, biological and clinical implications to investigate whether these unfavorable survival factors are common or unique in different individuals. In our work, we classify tumors with four EMT status, that is, EMT[low], EMT[mid], EMT[high]-NOS (Not Otherwise Specified), and EMT[high]-AKT (AKT pathway overactivation) subtypes. We find that EMT[high]-NOS subtype is driven by intrinsic somatic alterations. While, EMT[high]-AKT subtype is maintained by extrinsic cellular interplay between tumor cells and macrophages in an AKT-dependent manner. EMT[high]-AKT subtype is both unresectable and drug resistant while EMT[high]-NOS subtype can be treated with cell cycle related drugs. Importantly, AKT activation in EMT[high]-AKT not only enhances EMT process, but also contributes to the immunosuppressive microenvironment. By remodeling tumor immune-microenvironment by AKT inhibition, EMT[high]-AKT can be treated by immune checkpoint blockade therapies. Meanwhile, we develop TumorMT website (http://tumormt.neuroscience.org.cn/) to apply this EMT classification and provide reasonable therapeutic guidance.

[1] Department of Neurosurgery, Shengjing Hospital of China Medical University, Shenyang, Liaoning, China. [2] Department of Neurosurgery, The First Hospital of China Medical University, Shenyang, Liaoning, China. [3] Department of Neurosurgery, Chinese People's Liberation Army of China (PLA) General Hospital, Medical School of Chinese PLA, Institute of Neurosurgery of Chinese PLA, Beijing, China. [4]These authors contributed equally: Shuai Shen, Xing Liu, Qing Guo. ✉email: chengpengcmu@cmu.edu.cn; cmu071207@163.com; ahwu@cmu.edu.cn

Drug resistance and tumor reside continue to be the principal limiting factors for patient's survival[1,2]. Clinically, tumor residue and treatment resistance are always related to epithelial-mesenchymal transition (EMT)[3–5]. Unfortunately, most tumors undergo EMT during tumor development[3–5]. Following EMT, tumors exhibit high invasive ability[3] and unlimited proliferation[6] which is responsible for treatment resistance and high residual rate. For now, somatic alteration has been regarded as primary EMT driving mechanism and gives tumor cells aggressive phenotype[3,5,7,8]. For example, E-cadherin mutation could directly trigger EMT process[3,5]. Besides, copy number variation (CNV) loss like *BRCA1* could downregulate the expression of E-cadherin[7]. Also, over-activated oncogenic pathway like FOXD1 could promote EMT in glioma cells[8]. These somatic alterations are called intrinsic driving force of EMT. Recently, the tumor microenvironment (TME) factor has been recognized as an alternative EMT driving force. The interplay between tumor cells and non-tumor cells facilitates the EMT process of tumor cells and EMT-ed tumor cells shape TME into immunosuppressive subsequently[9,10]. Considering the heterogeneity of the driving force (genomic alteration or TME factor) in triggering EMT process, high EMT subtypes need to be established to investigate whether EMT-ed tumors process homogeneous or heterogeneous biological characteristics like overall survival time.

In this study, the heterogeneity of EMT status was firstly classified at the pan-cancer level through machine learning-based method. Four EMT subtypes (EMT$^{low}$, EMT$^{mid}$, EMT$^{high}$-NOS, and EMT$^{high}$-AKT) were recognized with distinct molecular, biological, and clinical features. We noticed that although with similar EMT enhancement, two mesenchymal like subtypes (EMT$^{high}$-NOS and EMT$^{high}$-AKT) were driven by intrinsic genomic alteration or extrinsic microenvironment interaction, respectively. EMT$^{high}$-NOS was characterized with extensive genetic alteration and could be treated by cell cycle targeting drugs. As for EMT$^{high}$-AKT subtype, EMT was driven by the interplay between tumor cell and macrophages through AKT pathway which is drug resistant and unresectable. Importantly, this interplay not only enhance EMT status, but also lead to the dysfunction of anti-tumor T cells which could be reverted by AKT inhibition. By remodeling tumor immune-microenvironment by AKT pathway inhibition, EMT$^{high}$-AKT could be treated by immune checkpoint blockade (ICB) therapies. Moreover, we introduced TumorMT (http://tumormt.neuroscience.org.cn) to conduct tumor EMT classification and provide reasonable therapeutic options for patients. Taken together, we established a novel EMT classification system, which would help us clarify the EMT driving mechanisms and provide valuable treatment guidance.

## Results

### Epithelial Mesenchymal Transition Core Gene (EMTCG) signature construction
To summarize the core genes associated with EMT process, we collected three well-established EMT related Gene Ontology (GO) gene sets from GSEA website (http://www.gsea-msigdb.org/gsea/index.jsp). Among these, one gene set (Go_Epithelial_to_Mesenchymal_Transition GO: 0001837) was annotated in positive correlation with EMT process, whereas the other two sets (Go_Mesenchymal_to_Epithelial_Transition GO: 0060231 and Go_Negative_Regulation_of_Epithelial_to_Mesenchymal_Transition GO: 0010719) represented the negative direction (Supplementary Data 1). By excluding genes participating in GO: 0060231 and GO: 0010719 from GO: 0001837, a total of 103 genes were filtered out as being positively correlated with EMT. Then, we conducted protein-protein interaction analysis using STRING website (https://www.string-db.org) and identified 58 genes exhibiting strong protein

interactions (Supplementary Data 2). Meanwhile, co-expression networks of the 103 genes were also depicted using COEXPEDIA website (https://www.coexpedia.org), eliciting 50 hub genes with substantial expression associations (Supplementary Data 2). Genes with protein and expression relationships were intersected to obtain a list of 35 genes, namely the EMT core genes (EMTCGs, Fig. 1a and Supplementary Data 2).

To test our signature's accuracy, we downloaded other 76 EMT related signatures from EMTome[11] (http://www.emtome.org). We calculated the correlation between our signature and these 76 signatures based on ssGSEA score. We found out that our signature was strongly positively correlated with these 76 signatures (mean correlation = 0.71, Std = 0.14, Supplementary Fig. 1a). Besides, we also calculated the correlation among each two signatures of 76 signatures and we found that the correlation (mean correlation = 0.71, Std = 0.14) between our signature and other 76 signatures is better than the correlation (mean correlation = 0.63, Std = 0.20) among each two signatures of 76 signatures (Supplementary Fig. 1b, c). Next, we compared the ssGSEA score of our EMTCG signature between mesenchymal and non-mesenchymal subtype tumors in four cancer types[12–15]. Firstly, we found out that our signature was significantly upregulated in mesenchymal glioma in four cohorts (Supplementary Fig. 1d). Secondly, we found out that our signature was significantly upregulated in mesenchymal colorectal cancer (CRC) in 10 cohorts (Supplementary Fig. 1e). Thirdly, we found out that our signature was significantly upregulated in mesenchymal breast cancer in two cohorts (Supplementary Fig. 1f). Finally, we found out that our signature was significantly upregulated in mesenchymal gastric cancer in six cohorts (Supplementary Fig. 1g). These results demonstrated the validity of our signature.

**EMT status could be disturbed due to multi-omics mechanisms.** Next, we tried to investigate the underlying molecular mechanisms that could affect EMT status. Firstly, genomic mutation frequency of the EMTCGs was compared to the baseline mutation frequency in each cancer type. In colon adenocarcinoma (COAD), rectum adenocarcinoma (READ), skin cutaneous melanoma (SKCM), and ovarian serous cystadenocarcinoma (OV), more EMTCGs were frequently mutant compared to other cancer types (frequency >5% and fold change >1.5) (Supplementary Fig. 2a and Supplementary Data 3). Secondly, exploration of the effect of CNVs on the abnormal expression of EMTCGs indicated that the upregulation of *EZH2, JAG1, SMAD2, and SMAD4* might be due to the amplified copy number of these genes (R > 0.3, frequency >30%, false discovery rate (FDR) < 0.0001) (Supplementary Fig. 2b and Supplementary Data 4). Finally, we found that reduced DNA methylation levels likely contributed to the upregulation of *ACVR1, BAMBI, ENG, S100A4, and TGFB3* (R < −0.3, FDR < 0.0001) (Supplementary Fig. 2c and Supplementary Data 5). Overall, 23.3% (196/840) cases were affected by genomic mechanisms, in which 5.6% (11/196) were mediated by multiple mechanisms (Fig. 1b). Taken together, these results offered a quantitative view of how different multi-omics mechanisms might disturb mesenchymal transition in tumor samples.

**EMT represents robust clinically relevant patterns and serves as an aggressive factor.** Subsequently we assessed prognostic implications of EMT based on EMTCG score (Supplementary Data 6). By univariate COX model, we found a significant negative correlation (hazard ratio (HR): 1.014–1.114, $p = 0.0118$) between the mesenchymal transition and OS times when 9415 samples were considered as a whole. Next, we conducted the

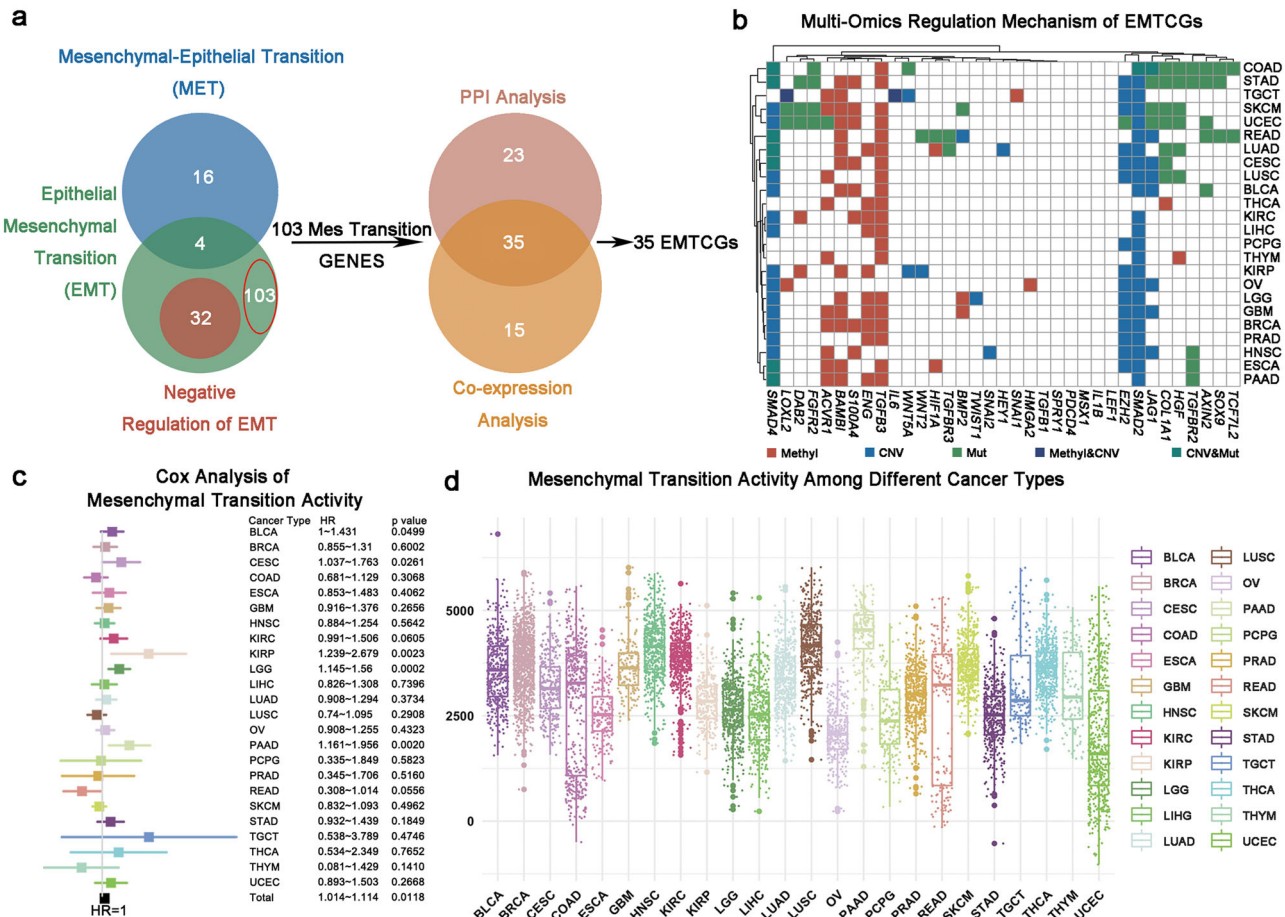

**Fig. 1 The landscape of EMTCG signature. a** Flowchart of construction of EMTCG signature. Left panel: Venn diagram shown that 103 genes involving only in the positive direction of mesenchymal transition were summarized. Right panel: Identification of 35 EMTCGs by protein-protein interaction network analysis and co-expression analysis. **b** Multi-omics regulation mechanism including somatic mutation, copy number variation (CNV) or DNA methylation of EMTCGs. Significant correlation ($p < 0.0001$) was represented on the heat-map. **c** Univariate Cox regression analysis access Hazard Ratio and prognostic significance of mesenchymal transition activity based on ssGSEA across different cancer types. **d** Boxplots of mesenchymal transition activity across 24 cancer types, as inferred using ssGSEA based on EMTCGs signature.

Cox regression analysis within each cancer type. Notably, in bladder urothelial carcinoma (BLCA), cervical squamous cell carcinoma and endocervical adenocarcinoma (CESC), kidney renal papillary cell carcinoma (KIRP), brain lower grade glioma (LGG), and pancreatic adenocarcinoma (PAAD), a significant negative correlation remained between mesenchymal transition and OS time (Fig. 1c and Supplementary Data 7). Furthermore, we also grouped patients based on the median EMTCG score. Higher mesenchymal transition status conferred poor prognosis at a pan-cancer scale ($p < 0.0001$), as did in BLCA, LGG, liver hepatocellular carcinoma (LIHC), and uterine corpus endometrial carcinoma (UCEC) (Supplementary Fig. 3). Among all the cancer types surveyed, the EMTCG score showed the most significant survival value in LGG (HR = 1.787, $p = 0.00035$).

As high EMT status could reduce patients' survival, we next investigated whether mesenchymal transition was consistently enhanced in tumors (Supplementary Fig. 4). First, the EMTCG score was compared between tumor samples and their paired normal samples. Although the EMT activity was enhanced in most tumor compartments, the mesenchymal transition activity was inhibited compared to paired normal samples in some tumor compartments. For example, in head and neck squamous cell carcinoma (HNSC), the mesenchymal transition activity levels of 81.4% (35/43) of the tumor samples were significantly enhanced,

whereas the levels were inhibited compared to normal tissues in 18.6% (8/43) of the samples. Next, we calculated the coefficient of variance (CV) of EMT activity within each cancer type to explore the fluctuation status. UCEC, READ, and COAD were found to exhibit high mesenchymal transition activity fluctuation (CV > 50%), whereas HNSC, lung squamous cell carcinoma (LUSC), and PAAD showed low mesenchymal transition activity fluctuation (CV < 20%). Besides, we also found a strong negative correlation (R = −0.682, $p = 0.0002$) between the mean activity and the CV, which indicated that in the case of general low mesenchymal transition tumor types, individuals with high mesenchymal transition activity could also be observed (Fig. 1d and Supplementary Data 8). Together, these findings indicated that the malignant mesenchymal transition was heterogeneous between tumor samples, highlighting the need for further studies to define different mesenchymal-like groups.

**EMT clustering system based on EMTCG.** To discriminate tumor samples according to mesenchymal transition status, unsupervised consensus cluster was used to classify 9415 samples across 24 cancer types based on transcriptomic expression of 35 genes consisting EMTCG. Following evaluation of cumulative distribution function, Delta area, and consensus cluster data, four

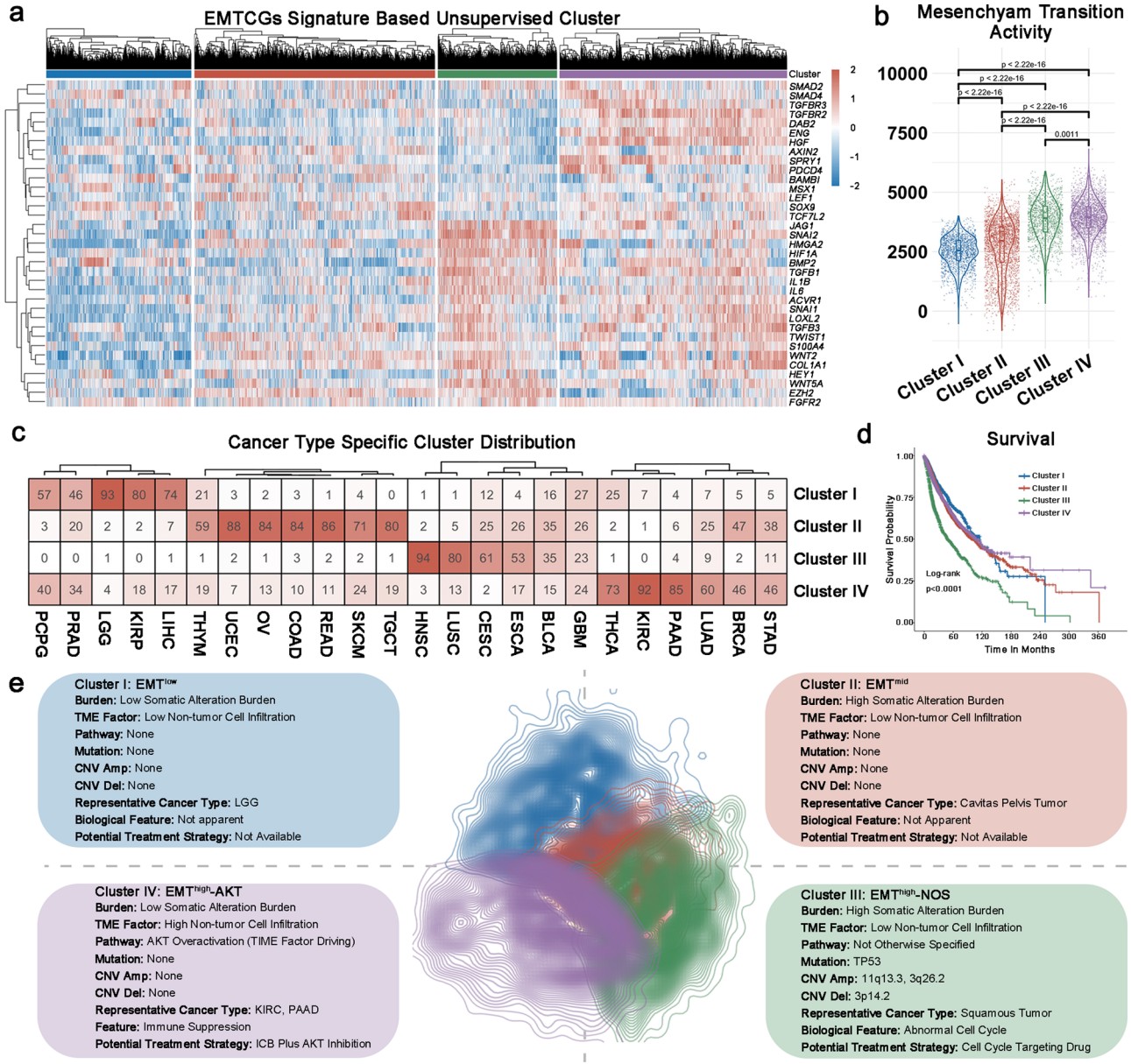

**Fig. 2 Epithelial origin tumors exhibit four EMT subtypes. a** Unsupervised consensus cluster of 9415 samples across 24 cancer-types based on 35 EMTCGs classified tumors into four subtypes. **b** Comparison of mesenchymal transition activity among four clusters. The mesenchymal transition activity was significantly up-regulated in Cluster III and IV; the Student's t test was used to analyze statistical significance; Cluster I, n = 1864; Cluster II, n = 3094; Cluster III, n = 1532; Cluster IV, n = 2925. **c** Distribution of individual cancer samples among four different EMT clusters. **d** Kaplan-Meier curve of overall survival of patients with different EMT subtypes for TCGA datasets; Cluster I, n = 1864; Cluster II, n = 3094; Cluster III, n = 1532; Cluster IV, n = 2925; Cluster III was statistically associated with unfavorable survival outcome. **e** Summarization of biological features among four EMT clusters.

EMT clusters were established (Fig. 2a and Supplementary Fig. 5a–d). PCA analysis revealed that before clustering the distribution of 24 cancer types were in chaos (Supplementary Fig. 5e) while consensus cluster could clearly classify tumor samples into four cluster (Supplementary Fig. 5f). Comparison of the EMTCG score among the four clusters indicated that Cluster I was characterized by an attenuated mesenchymal transition tendency (EMT^low), Cluster II by moderate mesenchymal transition activation (EMT^mid), Cluster III and IV exhibited enhanced mesenchymal transition profiles (EMT^high) (Fig. 2b). Since the sample number in each cancer type varied dramatically (for example, whereas 1102 cases of breast invasive carcinoma (BRCA) were available, only 120 cases were available for thymoma (THYM)), we sampled 10 patients of each cancer in each cluster and merged them together to eliminate the bias due to

cancer type specificity for 100 times[16]. For 100% (100/100) iterations, Cluster III and IV exhibited enhanced mesenchymal transition profiles (Supplementary Fig. 6a).

The clinical relevance of this mesenchymal transition classification was also investigated. Specifically, squamous carcinoma including HNSC, LUSC, CESC, and esophageal carcinoma (ESCA) tended to be enriched in Cluster III, whereas adenocarcinomas including thyroid carcinoma (THCA), PAAD, lung adenocarcinoma (LUAD), BRCA, and stomach adenocarcinoma (STAD) tended to be enriched in Cluster IV (Fig. 2c). Besides, the survival time was also compared, showing that Cluster III had the shortest survival (median survival = 48.6 months), whereas Cluster I, Cluster II, and Cluster IV exhibited relatively long survival (medial survival for Cluster I = 115.7 months, medial survival for Cluster II = 101.4 months, medial survival for Cluster

IV = 108.6 months) (Fig. 2d). To eliminate the bias due to cancer type specificity, we also introduced sampling strategy mentioned above. After sampling analysis, in 96% (96/100) iterations, Cluster III still had the shortest survival (Supplementary Fig. 6b). These analysis indicated two EMT$^{high}$ subtype might exist which urged us to investigate the heterogeneity between different EMT state tumors.

**Intrinsic and extrinsic features in EMT$^{low}$ and EMT$^{mid}$ tumors**. To explore potential mechanisms driving mesenchymal transition process, both intrinsic somatic alteration and extrinsic TME factor were explored for this purpose. The somatic mutation data (299 genes, which were considered as tumor driver mutations), somatic CNV data (84 loci, which were considered as tumor driver CNVs)[17,18], and tumor purity were analyzed among different EMT subtypes. In this part, all results were acquired with sampling analysis for 100 iterations.

In EMT$^{low}$ subtype (Fig. 2e), no characteristic mutation (Supplementary Data 9) or CNVs (Supplementary Data 10) were detected. From the extrinsic viewpoint (Supplementary Fig. 7a–c), these tumors had the lowest level of non-tumor cell infiltration for 100% (100/100) iterations.

In EMT$^{mid}$ subtype (Fig. 2e), no characteristic mutation (Supplementary Data 9) or CNVs (Supplementary Data 10) were detected. From the extrinsic perspective (Supplementary Fig. 7a–c), the EMT$^{mid}$ subtype exhibited a low fraction of non-tumor cells infiltration in 100% (100/100) iterations.

**Two EMT$^{high}$ subtypes are driven by distinct mechanisms**. Here, we established two mesenchymal like (EMT$^{high}$) subtypes with similar high EMT status. It should be important to explore the clinical, molecular and biological features between these two mesenchymal like subtypes. The clinical features between these two EMT$^{high}$ subtype was firstly investigated. Since the sample number in each clinical aspect varied drastically (for example, whereas 1102 cases of BRCA were available, only 120 cases were available for THYM), we made normalization of sample number to make sure our analysis does not be skewed by sample sizes as shown in Supplementary Data 11[16]. For example, in histological type part, the sample number of each cancer type was adjusted into 100 to calculate the sample number of every cancer type in each cluster. Firstly, we compared the difference of tumor composition fraction (Fig. 3a–c and Supplementary Fig. 8a, b). Cluster III was mainly composited of squamous carcinoma, including HNSC (24.3%), LUSC(20.8%), CESC (15.8%) and ESCA (13.7%). In total, these four cancer types composited 74.7% of Cluster III samples. As for Cluster IV type, top composition cancer types are all adenocarcinoma, including kidney renal clear cell carcinoma (KIRC, 13.3%), PAAD (12.4%) and THCA (10.55%). Secondly, we found that Cluster III was mainly composed of recurrent or primary samples while the portion distribution was relatively even in Cluster IV. Thirdly, we discovered that with histological grade progression, high EMT tumor samples tended to enrich in Cluster IV type (from Grade I to Grade IV: 15.4%, 18.5%, 17.7%, and 48.4%) instead of Cluster III type (from Grade I to Grade IV: 33.6%, 36.0%, 21.3%, and 9.1%). Lastly, age and gender distribution patterns were relatively similar between these two mesenchymal like subtypes. These results demonstrated there are substantial histopathological differences between Cluster III and IV with similar enhanced mesenchymal transition status.

Next, we tried to identify over-activated pathway in two EMT$^{high}$ subtypes. To do this, we first mapped the 35 genes in EMTCG to highlight their enrichment pathways using KEGG analysis. A total of 9 pathways were acquired (Supplementary Data 12), including well established pathway facilitating EMT

process, including TGF-β (norm $p = 1.5 \times 10^{-8}$), YAP (norm $p = 4.645 \times 10^{-12}$), and AKT (norm $p = 0.032$)[19,20]. Compared to Cluster I or Cluster II, all 9 pathways were over-activated in two EMT$^{high}$ subtypes. When pathway activity was compared between two mesenchymal like subtypes, we found that AKT pathway was over-activated in Cluster IV in 100% (100/100) iterations (Fig. 3d and Supplementary Fig. 9i). However, we didn't find specific over-activated pathway in Cluster III (Supplementary Fig. 9). Besides, we also classified cell lines ($n = 629$) from CCLE dataset into four EMT clusters (Supplementary Fig. 10a and Supplementary Data 13). Consistently, the AKT pathway were significantly enriched in Cluster IV (Supplementary Fig. 10b). To further demonstrate Cluster IV is specified with AKT pathway over-activation, a cohort of glioma samples were used for IHC analysis. The result indicated the phosphorylation level of AKT pathway was significantly upregulated in Cluster IV (Supplementary Fig. 11). Therefore, Cluster III was thus termed as the EMT$^{high}$-NOS (Not Otherwise Specified) subtype, whereas Cluster IV was termed as the EMT$^{high}$-AKT subtype.

Next, we attempted to further distinguish these two mesenchymal like subtypes to investigate the difference and similarity between them. To do this, we firstly performed different expression gene analysis between two EMT$^{high}$ subtypes using Limma R package and found out 548 genes were significantly upregulated in EMT$^{high}$-NOS subtype while 558 genes were significantly upregulated in EMT$^{high}$-AKT subtype (fold change >2, adjusted $p < 0.05$, Supplementary Data 14). Then, we performed GO analysis using genes upregulated in each mesenchymal like subtype separately. Genes in EMT$^{high}$-NOS subtype were mainly enriched in GO terms related to cell cycle (Fig. 3e and Supplementary Data 15). Genes in EMT$^{high}$-AKT subtype were mainly enriched in GO terms related to multicellular crosstalk (Fig. 3e and Supplementary Data 15).

Finally, we tried to explored the differential mesenchymal transition driving mechanisms between EMT$^{high}$-NOS and -AKT subtype. In EMT$^{high}$-NOS subtype, *TP53* was a recurrent mutation event in 86% (86/100) iterations (median OR = 2.20, 1.50–3.26). Compare to 27.52% mutation rate in EMT$^{high}$-AKT subtype, *TP53* mutation rate is as high as 65.78% in EMT$^{high}$-NOS subtype (Fig. 3f). Besides, recurrent deletion of 3p14.2 (in 86% (86/100) iterations, median OR = 1.65), and amplification of 3q26.2 (in 98% (98/100) iterations, median OR = 1.73) and amplification of 11q13.3 (in 97% (97/100) iterations, median OR = 1.85) are representative CNVs in EMT$^{high}$-NOS subtype. *TP53* mutation[21] and 11q13.3 amplification[22] explain why EMT$^{high}$-NOS subtype is featured with abnormal cell cycle. In EMT$^{high}$-AKT subtype, no characteristic mutation or CNV was identified. From extrinsic view, EMT$^{high}$-AKT subtype had the highest non-tumor cell infiltration in 100% (100/100) iterations. Besides, we found out that tumor purity negatively correlated with AKT pathway activation in EMT$^{high}$-AKT subtype, indicating the role of non-tumor cells in promoting EMT of this subtype (Supplementary Fig. 7a–d).

Taken together, we demonstrated that two mesenchymal like subtypes might be viewed as distinct EMT$^{high}$ subtypes in future clinical practice.

**Only EMT$^{high}$-AKT subtype but not EMT$^{high}$-NOS was resistant to targeting drugs and shown high residual tumor rate**. EMT was always related with drug resistance[20]. Thus, we tried to investigate drug responsiveness between four EMT subtypes using GDSC database[23]. After comparing the IC50 of different drugs among these clusters, we discovered that EMT$^{high}$-NOS was actually sensitive to targeting drugs (Supplementary Fig. 12a). Among 445 drugs compared, 129 drugs (29%) were sensitive,

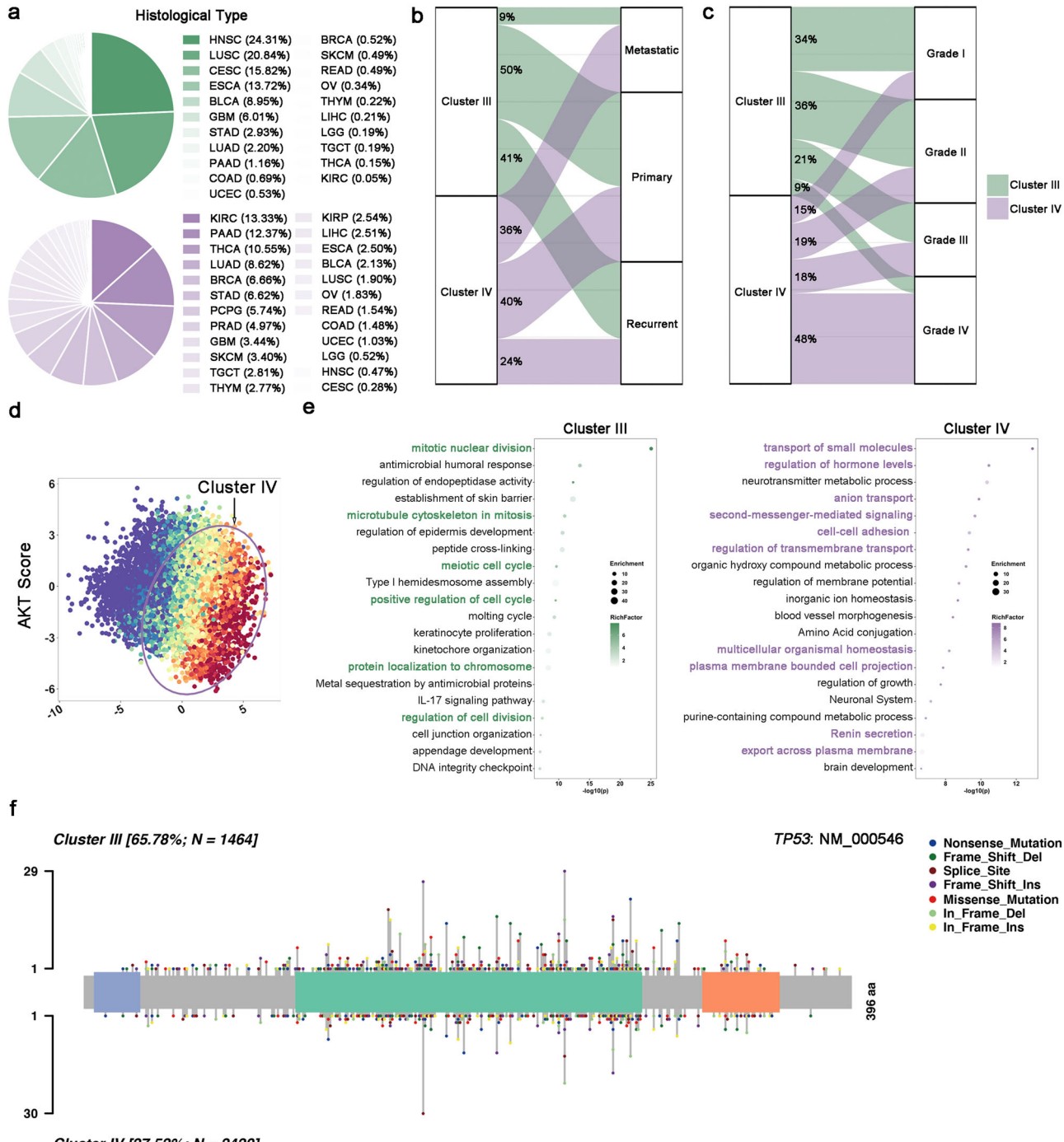

**Fig. 3 Multi-scale comparison between EMT<sup>high</sup>-NOS subtype and EMT<sup>high</sup>-AKT subtype. a** Histological type composition of EMT<sup>high</sup>-NOS subtype and EMT<sup>high</sup>-AKT subtype. EMT<sup>high</sup>-NOS subtype was mainly composited of squamous carcinoma while EMT<sup>high</sup>-AKT subtype was mainly composited of adenocarcinoma. **b** Sample portion composition of EMT<sup>high</sup>-NOS subtype and EMT<sup>high</sup>-AKT subtype. EMT<sup>high</sup>-NOS subtype was mainly composed of recurrent or primary samples while the portion distribution was relatively even in EMT<sup>high</sup>-AKT subtype. **c** WHO grade composition of EMT<sup>high</sup>-NOS subtype and EMT<sup>high</sup>-AKT subtype. With histological grade progression, mesenchymal transition tumor samples tended to enrich in EMT<sup>high</sup>-AKT subtype. **d** AKT pathway activity was significantly upregulated in EMT<sup>high</sup>-AKT subtype. **e** Gene ontology analysis of EMT<sup>high</sup>-NOS subtype and EMT<sup>high</sup>-AKT subtype. **f** Compared to EMT<sup>high</sup>-AKT subtype, the mutation rate of TP53 is significantly higher in EMT<sup>high</sup>-NOS subtype.

while only 6 (1%) were resistant in this subtype. Among 129 EMT<sup>high</sup>-NOS sensitive drugs, 50 (39%) drugs were related to cell proliferation, consistent with the abnormal cell cycle characteristic of EMT<sup>high</sup>-NOS subtype (Supplementary Fig. 12b). Therefore, although EMT<sup>high</sup>-NOS subtype had the worst survival status (Fig. 2d), it might be treated by targeting drugs in the future.

Next, although EMT<sup>high</sup>-AKT subtype had similar survival status to EMT<sup>low</sup> and EMT<sup>mid</sup> subtype, we compared the residual tumor rate to see whether EMT<sup>high</sup>-AKT subtype needed more non-surgical strategies compare to these low mesenchymal transition subtypes. The EMT<sup>high</sup>-AKT subtype had the highest residual tumor rate (19%, Supplementary Fig. 12c). Since EMT<sup>high</sup>-AKT subtype was characterized as AKT over-activation, we firstly

compared overall survival in this subtype between low AKT activation and high AKT activation samples. The survival status didn't alter between high AKT group and low AKT group in EMT^high-AKT subtype, indicating only inhibition of AKT pathway couldn't prolong survival in EMT^high-AKT subtype (Supplementary Fig. 12d). Besides, we also found that the IC50 of relative resistant drugs in EMT^high-AKT subtype were not correlated with AKT pathway activity, indicating targeting drugs couldn't be sensitized by AKT inhibition (Supplementary Fig. 12e, f). Taken together, these results indicating that EMT^high-AKT subtype was resistant to targeting drugs and other strategies should be developed.

**Only EMT^high-AKT subtypes is associated with immunosuppressive microenvironment.** EMT is consistently associated with impaired anti-tumor immunity[24]. We next assessed the immune reaction among the four subtypes. The *IFNG signature* was adopted as adaptive anti-tumor immunity marker[25], whereas the *TNF signature* as innate pro-inflammatory indicator[26]. Compared to EMT^low subtype, both IFNG activity and TNF activity were upregulated in the two EMT^high subtypes. However, when compared to EMT^mid subtype, the IFNG activity is similar between EMT^high-AKT subtype and EMT^mid subtype in 54% (54/100) iterations, whereas the IFNG activity is higher in EMT^high-NOS subtype than in EMT^mid subtype in 100% (100/100) iterations (Supplementary Fig. 13a). As for pro-inflammatory status, EMT^high-AKT subtype exhibits the highest TNF activity in 100% (100/100) iterations (Supplementary Fig. 13b). The unparallel innate and adoptive anti-tumor immunity in EMT^high-AKT subtype suggested that adaptive anti-tumor immunity might be inhibited in EMT^high-AKT subtype. To verify whether anti-tumor immunity was suppressed in the EMT^high-AKT subtype, we used gene set enrichment analysis (GSEA) to profile four immunosuppressive gene signatures. We found that these signatures were all enriched in the EMT^high-AKT subtype compared to non EMT^high-AKT subtype patients (Supplementary Fig. 13c).

Next, we further explored the immunosuppressive features in EMT^high-AKT patients. A lymphocyte signature[27] was applied to profile the CD8 + T cell fraction in the TME. Compared to other subtypes, the EMT^high-AKT subtype exhibited similar CD8 + T cell infiltration level in 97% (97/100) iterations (Supplementary Fig. 13d). The T cell dysfunction signature[25] was then adapted to evaluate the T cell function, which revealed that the EMT^high-AKT subtype presented the highest dysfunction score level in 100% (100/100) iterations (Supplementary Fig. 13e). As the major immune regulatory cells contributing to T cell dysfunction, the proportions of regulatory T cells (Tregs) and tumor associated macrophages (TAMs) were profiled among EMT subtypes. We found that TAM abundance and immunosuppressive M2-TAMs were significantly enriched in the EMT^high-AKT subtype compared to other EMT subtypes in 100% (100/100) iterations (Supplementary Fig. 13f), suggesting the important role of TAMs in facilitating immunosuppression in EMT^high-AKT subtype. As for Tregs, the abundance in EMT^high-AKT type was not significantly enhanced compared to that in EMT^mid or EMT^high-NOS subtype in 98% (98/100) iterations, suggesting Tregs might not participate in shaping immunosuppression in EMT^high-AKT subtype (Supplementary Fig. 13g).

**AKT activation dependent tumor-TAM feedback is important for shaping immunosuppression only in EMT^high-AKT subtype.** EMT related immunosuppression is always related with TAMs[9,28]. We next tried to investigate the role of AKT pathway in regulating the crosstalk between mesenchymal transition and TAM manipulation in different EMT subtypes. Breast cancer cell

lines and melanoma cell lines were used for this purpose, since breast cancer had the largest sample number ($n = 1102$) in TCGA cohort while melanoma is the first cancer type approved by FDA to use immune checkpoint blockade (ICB) therapy[29]. Four breast cancer cell lines (T-47D: EMT^low subtype, Luminal A; MDA-MB-468: EMT^mid subtype, basal like; HCC38: EMT^high-NOS subtype, claudin-low; MDA-MB-231: EMT^high-AKT subtype, claudin-low) and four melanoma cancer cell lines (MeWo: EMT^low subtype; A-375: EMT^mid subtype; SK-MEL-3: EMT^high-NOS subtype; WM-115 EMT^high-AKT subtype) were used for this purpose.

We firstly tested the mesenchymal transition status of tumor cells after being co-cultured with macrophages. Two mesenchymal markers, N-cadherin (N-cad) and Vimentin (VIM) were tested. N-cad and VIM were consistently upregulated in MDA-MB-231 cell after macrophage co-culturing. To test the role of the AKT pathway in this phenomenon, tumor cells were pre-treated with an AKT inhibitor (MK2206). MK2206 could only inhibited the facilitating mesenchymal transition promoting effects of TAMs on MDA-MB-231 cell line (Fig. 4a–d). These results indicated a specific mesenchymal transition facilitating role of TAMs on EMT^high-AKT subtype mediated by AKT pathway.

Next, we detected the role of different EMT subtypes on TAM infiltration alteration. Following co-culture with different subtypes of breast cancer cell lines, we detected the infiltration ability of macrophages. Although MDA-MB-468 (fold change 3.7) and HCC38 (fold change 2) also enhanced the infiltration of macrophages, MDA-MB-231 dramatically increased the infiltration (fold change 21.9). Besides, MK2206 could only abolish TAM promoting effect in the MDA-MB-231 cell line (Fig. 4a–d).

Furthermore, PD-L1, as major effector leading to the dysfunction of T cell, was analyzed on both tumor cells and macrophages. Using breast cancer cell line, we found THP-1 could only up-regulate expression of PD-L1 in MDA-MB-231 but not in other EMT subtype breast cancer cell through AKT pathway (Fig. 4e–h, upper panel). Additionally, MDA-MB-231 cells could up-regulate the PD-L1 expression of THP-1 more intensively than other EMT subtype breast cancer cell through AKT pathway (Fig. 4e–h, lower panel).

By using melanoma cell lines, similar tumor-TAM feedback loop was also observed in WM-115 cell line (EMT^high-AKT subtype, Supplementary Fig. 14). These results suggested that the AKT pathway was the specific hub mediating immunosuppression in EMT^high-AKT subtype by regulating the crosstalk between mesenchymal transition and TAM manipulation.

**AKT inhibition recover the immunosuppressive microenvironment of EMT^high-AKT tumor in vivo.** Next, we tried to test whether AKT signaling was responsible for immunosuppression of EMT^high-AKT subtype in vivo. To do this, we first conducted RNA-seq in three mouse cancer cell lines including mGSC (glioma), 4T1 (breast cancer) and B16-F10 (melanoma) to identify mouse EMT^high-AKT subtype tumor. Using three machine learning methods, all these three cell lines were classified into EMT^high-AKT subtype (Fig. 5a). Since mGSC was a spontaneous primary cancer cell line, which intimates more to primary human cancer than established cancer cell lines, mGSC was used for CyTOF analysis.

As indicated in Fig. 5b, an intracranial xenograft glioma model was established using mGSC cell line to harvest single-cell suspension for the CyTOF analysis with 37 antibodies (Supplementary Data 16). Firstly, we found that by inhibiting AKT signaling pathway, the immune cell infiltration tends to increase (Fig. 5c). Next, we gated into CD45+ immune cells and acquired 20 cell populations for further analysis (Fig. 5d). We observed a tendency of few macrophage infiltration in MK2206 group (Fig. 5e).

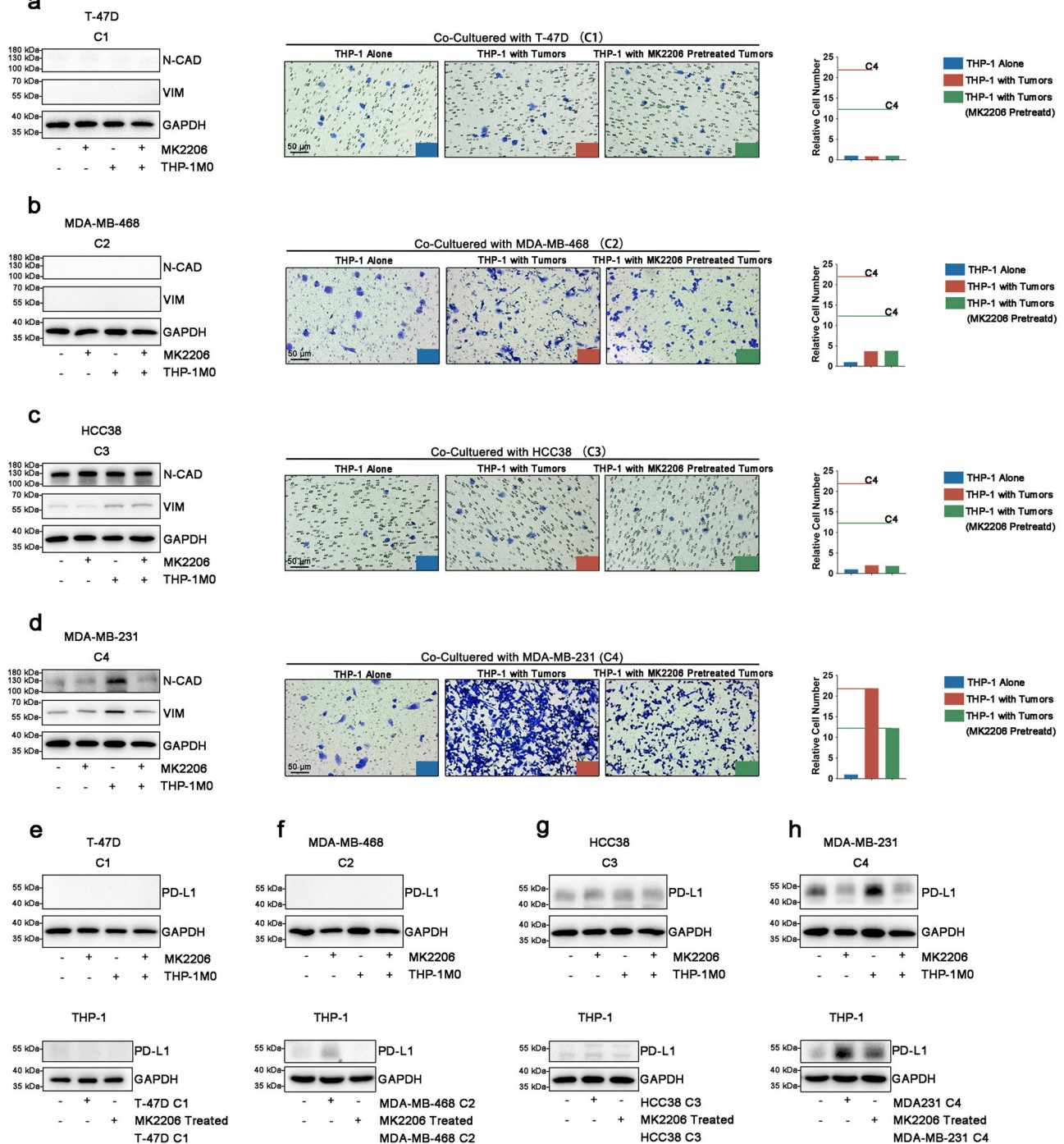

**Fig. 4 AKT pathway is important for mediating mesenchymal transition and immunosuppression only in EMT^high-AKT subtype. a** Left panel: Immunoblotting analysis of N-Cadherin and Vimentin indicated that macrophage could not promote mesenchymal transition of T-47D. Right panel: Transwell analysis indicated T-47D could not enhance infiltration ability of macrophages. **b** Left panel: Immunoblotting analysis of N-Cadherin and Vimentin indicated that macrophage could not promote mesenchymal transition of MDA-MB-468. Right panel: Transwell analysis indicated MDA-MB-468 enhanced infiltration ability of macrophages mildly and could not be inhibited by AKT inhibition. **c** Left panel: Immunoblotting analysis of N-Cadherin and Vimentin indicated that macrophage could partially enhance mesenchymal transition of HCC38 (Vimentin) and could not be inhibited by AKT inhibition. Right panel: Transwell analysis indicated HCC38 enhanced infiltration ability of macrophages mildly and could not be inhibited by AKT inhibition. **d** Left panel: Immunoblotting analysis of N-Cadherin and Vimentin indicated that macrophage could enhance mesenchymal transition of MDA-MB-231 which could be inhibited by AKT inhibition. Right panel: Transwell analysis indicated MDA-MB-231 enhanced infiltration ability of macrophages dramatically and could be inhibited by AKT inhibition. **e** PD-L1 expression did not up-regulate in co-cultured T-47D and macrophages. **f** PD-L1 expression did not up-regulate in co-cultured MDA-MB-468 and macrophages. **g** PD-L1 expression did not up-regulate in co-cultured HCC38 and macrophages. **h** PD-L1 expression up-regulated in co-cultured MDA-MB-231 and macrophages which could be inhibited by AKT inhibition. For all transwell images in this figure, scale bar = 50 μm.

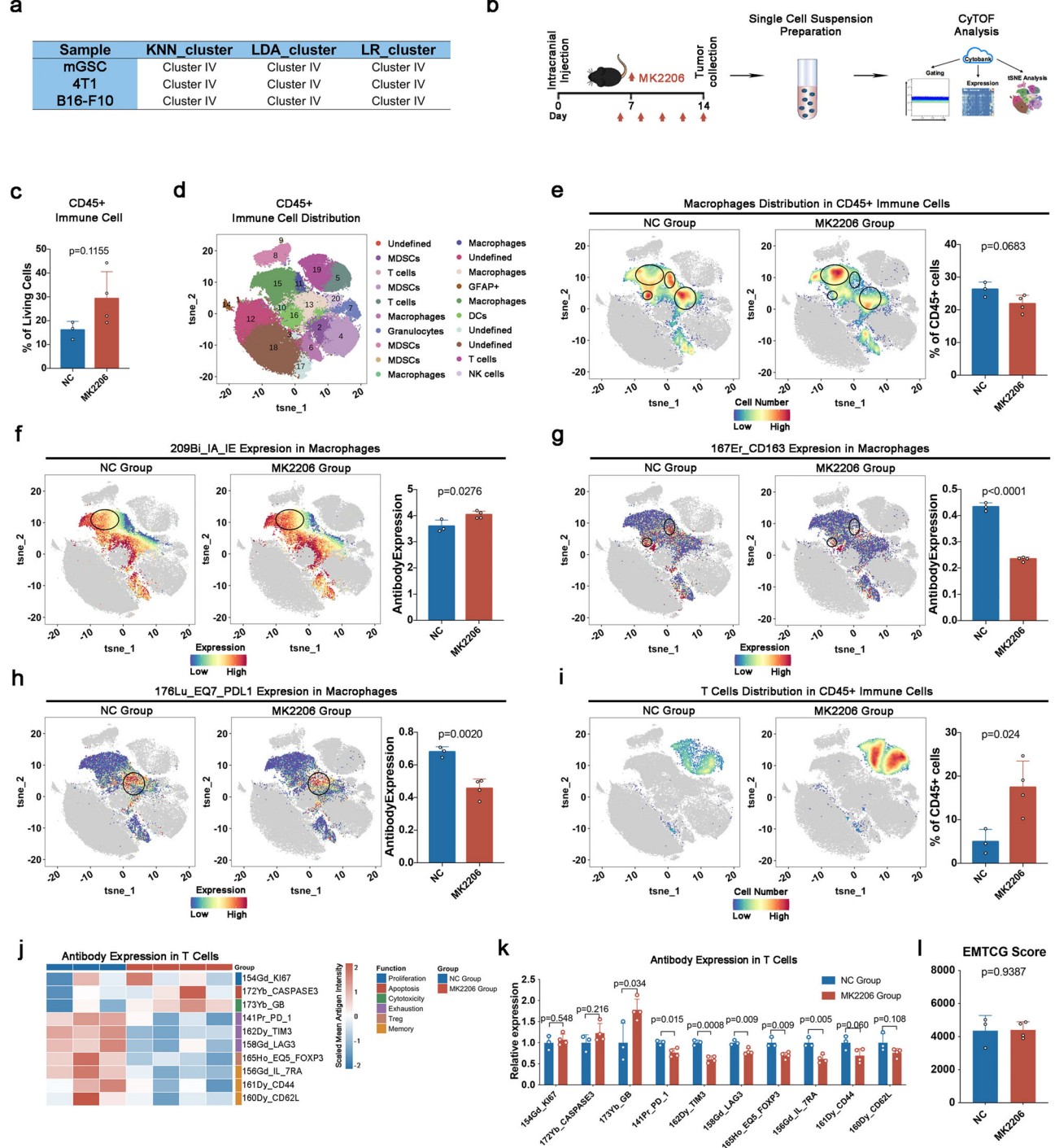

**Fig. 5 AKT inhibition could reverse the immune-suppressive phenotype of the EMT^high-AKT subtype in vivo. a** EMT classification of mGSC, 4T1 and B16-F10 cell line using machine learning method. **b** Flowchart of CyTOF Analysis. **c** Comparison of CD45+ immune cell's percentage between NC group and MK2206 group. **d** 20 clusters were acquired in CD45+ immune cells. **e** TAM's infiltration has a tendency to decrease in MK2206 group compared to NC group. **f** IA_IE expression is upregulated in TAM's of MK2206 group. **g** CD163 expression is downregulated in TAM's of MK2206 group. **h** PDL1 expression is downregulated in TAM's of MK2206 group. **i** T cell's infiltration increased in MK2206 group compared to NC group. **j** Expression heatmap of T cell's function markers. **k** T cell's dysfunction status was reversed by MK2206 without increase of memory T cells. **l** Mesenchymal transition status did not alter after MK2206 inhibition. Error bars indicate SD; the Student's t test was used to analyze statistical significance between 2 groups; In NC group, n = 3 biological replicates; In MK2206 group, n = 4 biological replicates.

Importantly, macrophage distribution pattern (illustrated by circles in Fig. 5e) was different between NC and MK2206 groups, indicating macrophage function might differ between these two groups. IA_IE (M1), CD163 (M2), and PDL1 were used to evaluate macrophage functional phenotype. The expression of IA_IE was upregulated in the MK2206 group, while CD163 and PDL1 were downregulated in MK2206 group (Fig. 5f–h).

To further explore the functional status of TAMs, mRNA expression level of TAM's proinflammatory (M1) or anti-inflammatory (M2) markers were also detected using RT-qPCR

test. As shown in Supplementary Fig. 15, M1 markers including Ifng (Supplementary Fig. 15a), Tnfa (Supplementary Fig. 15b) and Cd86 (Supplementary Fig. 15c) didn't alter while M2 markers including Cd206 (Supplementary Fig. 15d), Arg1 (Supplementary Fig. 15e) and Trem2 (Supplementary Fig. 15f) downregulated in mGSC xenograft model after MK2206 administration. These results indicated that AKT pathway inhibition could block M2 transformation of macrophages.

Next, we investigated whether inhibiting AKT pathway was helpful for antitumor effects. Compared to the NC group, more T cells infiltrated into tumor tissue in MK2206 group (Fig. 5i). The cytotoxic ability (GB) of T cell was increased in the MK2206 group (Fig. 5j, k). T cell exhaustion markers (PD_1, TIM3, and LAG3) were downregulated in the MK2206 group (Fig. 5j, k). These results indicated that T cell dysfunction was reversed by AKT inhibition. Meanwhile, the proportion of peripheral macrophages, microglia, dendritic cells, NK cells, and MDSCs were also investigated. More dendritic cells were infiltrated into the MK2206 group (Supplementary Fig. 16a–d).

To rule out the possibility that AKT inhibition induced cancer cell differentiating into a more epithelial state to recover immunosuppressive microenvironment, mGSC xenograft tissues were sequenced. The result shown MK2206 didn't alter the mesenchymal status of cancer cell (Fig. 5l). Taken together, these findings demonstrated that AKT signaling inhibition could reverse the immunosuppressive phenotype of the EMT$^{high}$-AKT subtype in vivo.

**Immune checkpoint blockade plus TME remodeling by AKT inhibition could prolong survival in EMT$^{high}$-AKT subtype.** Considering the immunosuppression in EMT$^{high}$-AKT subtype, we tried to test whether ICB therapy would work for these patients. Firstly, we investigated whether single ICB would prolong survival in this subtype. Four ICB clinical trial cohorts were used to do this. In Imvigor210 dataset[30], one-way ANOVA analysis revealed EMT$^{high}$-AKT subtype exhibiting the lowest response fraction (17%, $p = 0.0053$, Fig. 6a). Besides, t test revealed that, ICB response rate in EMT$^{high}$-AKT subtype was dependent on AKT pathway activity (Fig. 6b). In GSE78220[31], although patients were not classified into all 4 EMT subtypes due to limited sample numbers, we found that response rate was also dependent on AKT pathway activity (Fig. 6c). Next, two anti-CTLA4 trials (phs000452[32] and SRP067586[33]) were combined together to test the response rate of anti-CTLA4 therapy. Consistently, EMT$^{high}$-AKT subtype exhibited the lowest response fraction (26%, $p = 0.0056$, Fig. 6d). Besides these trials, we applied the TIDE algorithm[25] to evaluate the immunotherapeutic response and compared the response rate between the four EMT subtypes in TCGA dataset. Similarly, the EMT$^{high}$-AKT subtype exhibited the lowest fraction of response (22%) dependent on AKT pathway activity (Fig. 6e, f). Then, we applied 4T1, B16-F10 and mGSC xenograft model to test the ICB therapy response. Single ICB administration could not reduce tumor burden in 4T1, B16-F10 and mGSC cell lines and could not prolong survival in B16-F10 and mGSC cell lines (Fig. 6g–o). For 4T1, no death due to tumor was recorded under the observation period. These findings suggested that EMT$^{high}$-AKT subtype was resistance to singe administration of immunotherapy.

Since the immunosuppressive microenvironment of EMT$^{high}$-AKT subtype was shape by AKT pathway, we next investigated whether AKT inhibition could enhance the treatment value of ICB therapy in EMT$^{high}$-AKT subtype cell lines. The drug administration strategy was illustrated in Supplementary Fig. 17. Combination of ICBs (including anti-PD-L1, anti-PD-1 and anti-CTLA-4) and MK2206 significantly inhibited tumor growth

in 4T1, B16-F10 and mGSC cell lines and prolonged survival in B16-F10 and mGSC cell lines (Fig. 6g–o). For 4T1, no death due to tumor was recorded under the observation period. Besides, MK2206 did not cause apparent structural alteration or toxicity in liver or kidney (Supplementary Fig. 18). Therefore, AKT inhibition relieved the immunosuppression and therapeutic resistance to ICB treatment in EMT$^{high}$-AKT subtype.

**TumorMT: A comprehensive resource for exploring tumor mesenchymal transition status.** Our results highlighted the necessity for precisely identifying mesenchymal transition status among patient samples (Supplementary Fig. 19). Therefore, we developed the TumorMT website (http://tumormt.neuroscience.org.cn) (Supplementary Fig. 20a–e). The in-house glioma dataset, Chinese Glioma Genome Atlas (CGGA), was used as a paradigm and validation of the EMT subtype assignment accuracy and practicability of the website. The website classified the 388 Glioblastoma Multiforme (GBM) samples of CGGA into four EMT clusters (Supplementary Fig. 21a–c). Consistent with the TCGA cohort, EMT$^{high}$-AKT subtype GBM samples in the CGGA cohort exhibited the lowest tumor purity level and the highest level of non-tumor compartment infiltration (Supplementary Fig. 21d–f). Moreover, anti-tumor immunity was impaired in this subtype owing to high M2-TAM infiltration, which led to T cell dysfunction (Supplementary Fig. 21g, h). These results demonstrated the accuracy of our findings and the practicability of the TumorMT website.

## Discussion

For now, several studies have established reasonable clustering system for pan-cancer patients in the field of epithelial mesenchymal transition[34,35]. However, how to discern EMT driving force from intrinsic and extrinsic perspectives remains challenging and raised questions regarding the true role of EMT in tumor biology[36]. In this study, we generated an EMTCG signature that robustly classified tumors into four EMT subtypes. Especially, two EMT$^{high}$ subtypes were found to be driven by either genomic alteration or TAM crosstalk which spurred us to explore the heterogeneity in high EMT status separately. The unresectable, drug resistant EMT$^{high}$-AKT subtype was immunosuppressive and could be treated by combination of blocking ATK pathway and immune checkpoint blockade.

Routinely, two approaches have been applied to evaluate tumor EMT status[5,6,8,10,34,35]. The first approach incorporates experimental methods to detect the expression of individual mesenchymal markers[8,10]. The second approach utilizes score-based bioinformatic approaches to detect the overall enrichment of EMT genes[34,35]. However, these methods have intrinsic limitations. On one hand, as EMT is a multi-mechanism hybrid process, using a single experimental marker might introduce selection bias into the evaluation. On the other hand, score-based bioinformatic approaches evaluate EMT activity by comparing the score within the same cohort and might lose efficacy when applied toward independent clinical patient. Our work solved these problems to some degree. Firstly, our EMTCG signature was constructed by using co-expression analysis and protein-protein-interaction analysis to include all key genes in EMT process and thus avoid marker selection bias. Secondly, by using unsupervised consensus cluster, the characteristic cluster center of the four EMT subtypes could be acquired. By calculating the similarity distances to the cluster centers of the four EMT subtypes, individual clinical patient could also get categorical information regarding to our classification. Moreover, we developed TumorMT website, which may provide precise classification and treatment information based on our findings.

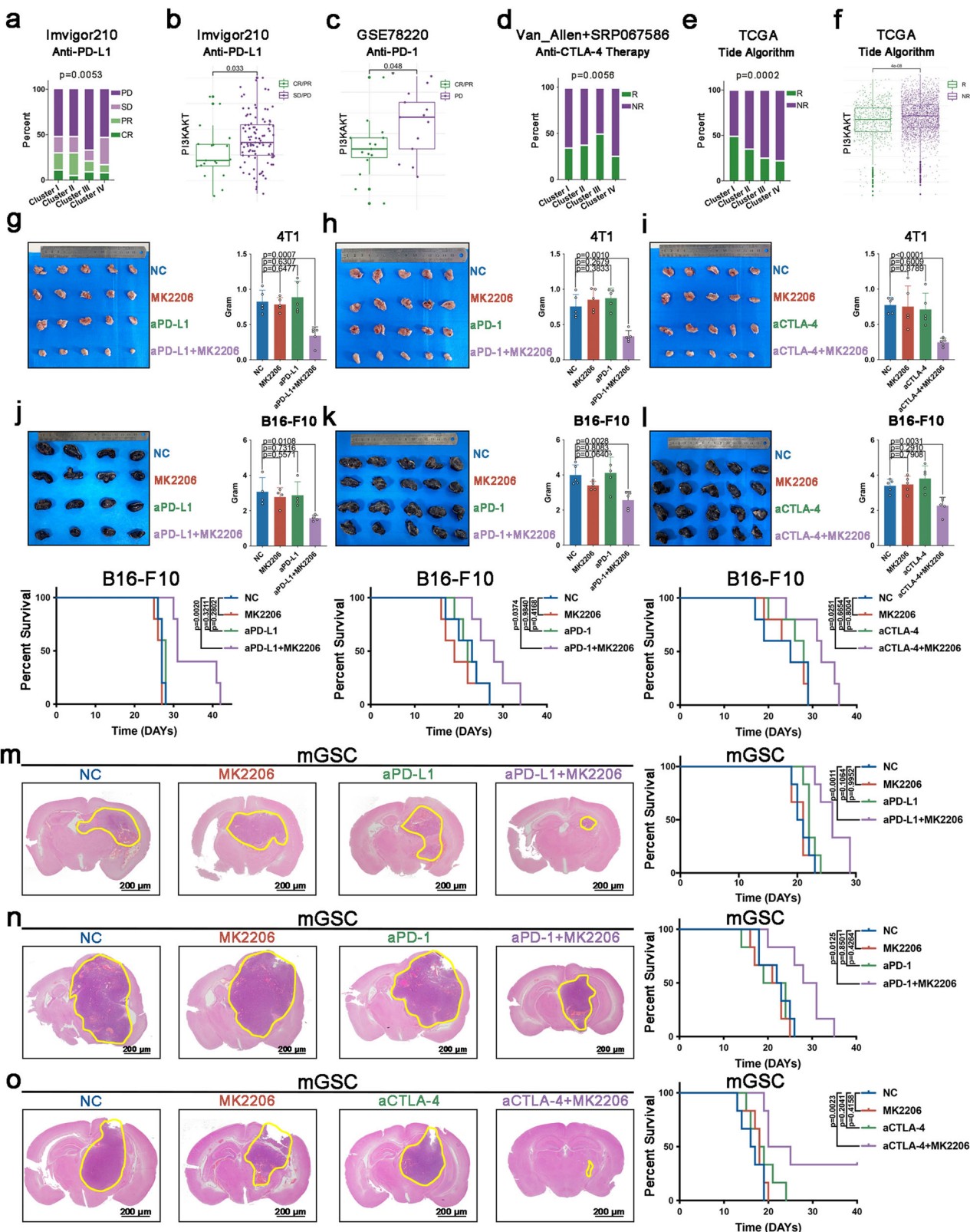

Although a lot of tumors have been featured by enhanced mesenchymal phenotypes, we still have few knowledge about the heterogeneity among these patients[36]. Especially, although both genetic alteration and TME could promote mesenchymal transition, existing methods are not be able to identify dominant driving mechanism in individual mesenchymal-like samples which hinders in choosing the most proper EMT targeting treatment strategy[36]. Our work established two mesenchymal like subtypes (EMT$^{high}$-NOS and EMT$^{high}$-AKT), which were found to have distinct biological and clinical features. Compared to EMT$^{high}$-NOS subtype, EMT$^{high}$-AKT is drug resistant, and unresectable. Importantly, our findings suggested that the EMT$^{high}$-NOS subtype was driven by intrinsic somatic alterations which could be treated by cell cycle targeting strategies, whereas

**Fig. 6 ICB plus AKT inhibition could prolong survival in EMT^high^-AKT subtype. a** Comparison of anti-PD-L1 therapy response rate among the four EMT clusters in IMvigor210 cohort; Complete response (CR), Partial response (PR), Stable disease (SD), Progressive disease (PD). **b** Comparison of AKT pathway activity in EMT^high^-AKT subtype patients with different immunotherapy responses in IMvigor210 cohort; CR/PR group, $n = 19$; SD/PD group, $n = 95$. **c** Comparison of AKT pathway activity with different immunotherapy responses using anti-PD-1 in GSE78220 cohort; CR/PR group, $n = 15$; PD group, $n = 13$. **d** Comparison of anti-CTLA-4 therapy response rate among the four EMT clusters in Van_allen+SRP067586 cohort; Response (R), Nonresponse (NR). **e** Comparison of ICB therapy response rate among the four EMT clusters in TCGA cohort. Response (R) and non-response (NR). **f** Comparison of AKT pathway activity in EMT^high^-AKT subtype patients with different immunotherapy responses in TCGA cohort; R group, $n = 646$; NR group, $n = 2278$. **g** MK2206 could overcome anti-PD-L1 resistance in 4T1 xenograft tumors; $n = 5$ biological replicates. **h** MK2206 could overcome anti-PD-1 resistance in 4T1 xenograft tumors; $n = 5$ biological replicates. **i** MK2206 could overcome anti-CTLA-4 resistance in 4T1 xenograft tumors; $n = 5$ biological replicates. **j** MK2206 could overcome anti-PD-L1 resistance in B16-F10 xenograft tumors; $n = 4$ biological replicates for tumor burden assay; $n = 5$ biological replicates for survival assay. **k** MK2206 could overcome anti-PD-1 resistance in B16-F10 xenograft tumors; $n = 5$ biological replicates. **l** MK2206 could overcome anti-CTLA-4 resistance in B16-F10 xenograft tumors; $n = 5$ biological replicates. **m** MK2206 could overcome anti-PD-L1 resistance in mGSC intracranial xenograft tumors; $n = 6$ biological replicates. **n** MK2206 could overcome anti-PD-1 resistance in mGSC intracranial xenograft tumors; $n = 6$ biological replicates. **o** MK2206 could overcome anti-CTLA-4 resistance in mGSC intracranial xenograft tumors; $n = 6$ biological replicates. Error bars indicate SD; the Student's $t$ test or Kaplan-Meier analysis was used to analyze statistical significance between 2 groups; For (**m–o**), scale bar = 200 μm.

the EMT^high^-AKT subtype was driven by the crosstalk between tumor and TME which could be treated by ICB combined with TME remodeling. In the EMT^high^-NOS subtype, somatic alterations occurred in cell cycle-related factors such as *TP53 and Cyclin D1*. Their abnormality might cause tumor cells to undergo devastating cellular mitosis. In the EMT^high^-AKT subtype, M2-TAMs comprised the representative compartments in TME and promoted mesenchymal transition. In EMT^high^-AKT subtype, microenvironment factors, especially TAMs, weighs more in shaping mesenchymal transition than genetic variation factors. Therefore, intrinsic and extrinsic mesenchymal transition enhancing mechanisms might lead to distinct biological and clinical outcome, which need to be managed differently.

Increasing evidences have suggested that there are complex interactions between tumor mesenchymal transition and immunosuppressive phenotype. Here, we observed a specific AKT pathway depending tumor cell and TAM interaction facilitating both mesenchymal transition and immunosuppression in EMT^high^-AKT subtype tumors. Marks et al.[37] also demonstrated AKT inhibition was related with favorable immune profile, which was consistent with our CyTOF detection results. However, as was in Fig. 6g–o, AKT inhibition could not prolong overall survival of tumor patients. Our CyTOF analysis answered why AKT inhibition could not prolong overall survival to some extent. Although AKT inhibition reverted immunosuppressive status of TAMs and increased cytotoxic ability of T cells, AKT inhibition also tended to decrease the ration of memory T cells. On the other hand, ICBs have been demonstrated to increase the number ration of memory T cells[38]. However, ICBs didn't work in EMT^high^-AKT subtype, since the substantial tumor-TAM interaction significantly upregulated the expression of immune checkpoints in EMT^high^-AKT subtype (Fig. 4e–h). Previous studies have demonstrated that ICBs only work to the status that overexpression of immune checkpoints were due to T cell cytotoxicity[39,40]. Blocking non-CD8 + T cell origin-induced PD-L1 expression could help to enhance response rate of ICB therapy[41]. As shown in Fig. 4e–h, AKT inhibition could decrease non-CD8 + T cell origin-induced PD-L1 expression which indicated to increase ICB response rates. Here, by using three cell models and three ICB regimens, we demonstrated the universe feasibility of ICB treatment plus AKT inhibition. In this strategy, AKT inhibition was used to block PD-L1 overexpression and to revert immune suppressive microenvironment while ICBs were used to increase the ration of memory T cells.

Our findings highlighted the distinct molecular, biological and clinical features among patients with enhanced mesenchymal transition, indicating that different treatment strategies should be applied.

## Methods

**Data acquisition**. TCGA multi-omics data (clinical information '2018-09-13 version', CNV profile '2016-08-16 version', gene mutation profile '2016-12-29 version', DNA methylation profile 'Methylation450K 2016-12-29 version', transcriptomic expression profile '2016-12-29 version, RNA-seq', and reverse-phase protein array profile '2016-08-16 version') were downloaded from the Xena Website (https://xenabrowser.net/). The cancer cell line transcriptomic expression profile '20180929 version, RNA seq' was downloaded from Cancer Cell Line Encyclopedia (CCLE) Website (https://portals.broadinstitute.org/ccle/). Colorectal cancer datasets (KFSYSCC cohort, FRENCH cohort, GSE2109 cohort, GSE37892 cohort, GSE35896 cohort, GSE23878 cohort, GSE20916 cohort, GSE17536 cohort, and GSE13067 cohort; Transcriptome) were downloaded as the authors indicated[13]. Breast cancer dataset (FUSCCTNBC cohort; Transcriptome) was downloaded as the authors indicated[14]. Gastric cancer datasets (ACRG cohort, KUCM cohort, KUGH cohort, MDACC cohort and SMC cohort; Transcriptome) were downloaded as the author indicated[15]. IMvigor210CoreBiologies data (RNA seq) were downloaded using the R package (version 1.0.0) provided by the following website (http://research-pub.gene.com/IMvigor210Core Biologies/packageVersions/). GSE78220 (RNA-seq) data was downloaded from GEO website; Anti-CTLA4 clinical trial data (RNA seq) was acquired from dbGaP: phs000452 (https:// github.com/vanallenlab/VanAllen_CTLA4_Science_RNASeq_ TPM/ commit/3d1793629716cc1fd8e7334ea3bf593a20e6fe07)[32] and SRA: SRP067586 cohort[33]. We used pre-processed data provided by these authors. The details of the sample information included in this work are summarized in Supplementary Data 17.

**CGGA patient samples**. A total of 388 GBM samples were included in our study[42,43]. These samples were acquired by CGGA. Each sample was diagnosed and independently confirmed by two neuropathologists based on the 2007 WHO classification guidelines[44]. Only samples consisting of > 80% tumor cells were selected for transcriptomic profiling (RNA-seq). The details of the sample information included in this work are summarized in Supplementary Data 17. The CGGA cohort data could be obtained from the CGGA database (http://www.cgga.org.cn).

**Wu lab glioma samples**. Glioma tissue samples from patients were acquired from our institution with the assistance of neuronavigation between October 10, 2020 and August 18, 2022. Samples were sent for RNA-seq and classified into four EMT subtypes according to transcriptomic profile. 19 samples with sufficient tissue block were sent for immunohistochemistry

staining. The sample information of Wu lab could be found in Supplementary Data 17.

**Antibodies and reagents**. Detailed information on the antibodies and reagents used in this study is listed in Supplementary Data 18.

**PPI analysis**. PPI was performed using String Website (https://string-db.org/). Gene ID was used as input data. The interaction network using combined score was then imported into Cytoscape 'version 3.6.1' (https://cytoscape.org) for network analysis. The genes that had a degree (K) within the top two-thirds of all genes were considered node genes.

**Co-expression analysis**. Co-expression analysis was performed using the Coexpedia Website (http://www.coexpedia.org/). Gene ID was used as input data. The genes with a score within the top two-thirds of all genes were considered co-expression genes.

**Mutation analysis**. The baseline mutation frequency in each cancer type was calculated first (Baseline mutation frequency is defined as the mutation rate of the whole genome in a certain cancer type by calculating the ratio between positive mutation event number and total observed event number). We then calculated the ratio between mutation frequency of EMTCGs with the baseline mutation frequency as fold change in each cancer type. A mutation frequency >3% and fold change >1.5 was considered significant.

**DNA methylation analysis**. R software 'version 3.5.1' was used to analyze the Pearson correlation between the expression of EMTCGs and DNA methylation level (β-value). Three criteria were used to identify the abnormal expression of EMTCGs affected by DNA methylation for each cancer type: [1] the probe with the largest average β value was taken into account; [2] Pearson correlation between expression and DNA methylation probe < -0.3; [3] FDR < 0.05 was considered statistically significant. FDR adjust was done using Benjamin and Hochberg's method[45].

**CNV analysis**. CNV values were acquired by the Xena Website using TCGA FIREHOSE pipeline based on GISTIC2. R software 'version 3.5.1' was used to analyze the Pearson correlation between the expression of EMTCGs and CNV. Three criteria were used to identify the abnormal expression of EMTCGs affected by CNV for each cancer type: [1] More than 30% of the tumor samples had the amplification or deletion of loci; [2] Pearson correlation between expression and CNV > 0.3; [3] FDR < 0.05 was considered statistically significant. FDR adjust was done using Benjamin and Hochberg's method[45].

**Mesenchymal transition activity evaluation**. Mesenchymal transition activity of individual samples was evaluated using the ssGSEA method supported by GSVA R package 'version 1.28.0'[46]. The accuracy of ssGSEA to predict pathway activity has already been validated by other work[47]. Transcriptomic expression profile was used as input "expr" for the analysis and EMTCG signature was used as input "gset.idx.list" for the analysis. Following parameters are used: method=ssgsea, kcdf=Gaussian, abs.ranking=FALSE.

**Survival analysis**. Univariate Cox regression analysis was chosen to assess the hazard ratio of mesenchymal transition. 'coxph' function in 'survival' R package 'version 3.1-11' was used to do

univariate cox regression analysis. Survival time is used as time value, survival status is used as status and ssGSEA score of EMTCG signature is used as covariate[47]. HR > 1 was considered a harm factor, while HR < 1 was considered a protective factor. $p < 0.05$ was considered statistically significant.

**Machine learning based cluster analysis**. The ConsensusClusterPlus R package 'version 1.46.0' was used for K-Means cluster analysis of the TCGA cohort[48]. The expression profile of 35 EMTCGs was first normalized using the scale function of R software and then used as the input for cluster analysis. Machine learning parameters were as follows: Permutations = 100, Distance = Euclidean, InnerLinkage = Average, FinalLinkage = Average, and ClusterAlg = km.

**Sample number normalization between different clusters**. The number of samples for each categorical variable (histological type, sample portion, WHO grade, age and gender) were normalized into 100[16]. We then grouped samples into each EMT subtype.

**Sampling analysis**. To make sure our analysis didn't skew by sample number variation, we sampled 10 patients of each cancer in each cluster and merged them together to eliminate the bias due to cancer type specificity. Sampling would be repeated for 100 times to remove false positive results.

**KEGG analysis**. KEGG analysis was performed using the ClueGo plug-in unit 'version 2.5.3' provided by Cytoscape. Medium was chosen as Network Specificity. Two-sided hypergeometric test was chosen as statistical method. Bonferroni step down was chosen for p value correction and an FDR < 0.05 was considered statistically significant. FDR was done by Cytoscape automatically.

**Removal of batch effect**. The ComBat function of the SVA R package 'version 3.30.1' was used to remove the batch effect between different data cohorts[49]. Cohort variable was considered as batch variable. Covariable (mod parameter) was not considered in this analysis. Par.prior parameter was set as true. PCA analysis was used to test the result of removal of the batch effect.

**Identification of EMT Subtypes of Non-TCGA Samples**. Identification of non-TCGA samples was done with following steps. For human samples: (1) The batch effect between external cohorts and TCGA cohort was removed; (2) Each individual sample in external cohort was scaled with TCGA cohort separately; (3) Euclidian distance between individual samples and four TCGA EMT cluster centers was calculated using the expression of 35 MTCG genes. Euclidian distance$=\sqrt{\varepsilon(nonTCGAi - TCGAi)^2}$, i = 35, ε means summation; (4) Individual samples were classified into the corresponding subtype with the shortest distance.

For mouse samples: (1) The batch effect between external cohorts and TCGA cohort was removed; (2) three machine learning method, K-NearestNeighbor (KNN), Linear Discriminant Analysis (LDA) and Logistic Regression (LR) were used to identify samples into relative clusters. The sequencing data of mGSC (mouse spontaneous glioma cell) and 4T1 were acquired using xenograft tissue from our group and the sequencing data of B16-F10 was acquired from GSE174724.

**Identification of driver mutations**. The driver mutation list was downloaded from Bailey et al.[18]. The Maftools R package 'version 1.8.10' was used to determine driver mutations in each EMT subtype. Mutations with adjPval <0.05 and fold change >1.5 were

considered driver mutations. adjPval was done by Maftools R package automatically.

**Identification of driver copy number variations**. The driver CNV list was downloaded from Hoadley et al.[17]. The chi-square test was used to determine driver mutations in each EMT subtype. CNV with FDR < 0.05 and fold change >1.5 were considered as driver CNV. FDR adjust was done using Benjamin and Hochberg's method[45].

**Tumor purity analysis**. Stromal and immune scores were calculated using the ESTIMATE R package 'version 1.0.13' (https://sourceforge.net/projects/estimateproject/), and tumor purity was estimated according to this formula:purity = COS $(0.6049872018 + 0.0001467884 * estimatescore)$[28].

**IFNG activity and TNFA activity calculation**. IFNG activity was acquired by calculating the geometric mean of marker genes defined by Jiang et al.'s work[25]. TNFA activity was acquired using the ssGSEA method using marker genes defined by Wei et al.'s work[26].

**GSEA analysis**. Gene Set Enrichment Analysis (GSEA, http://www.broadinstitute.org/gsea/index.jsp, version 4.3.2) was performed to explore whether the identified sets of genes showed statistical differences between the two groups using default parameters[50]. Expression profile was used as expression dataset set. EMT[high]-AKT group or non-EMT[high]-AKT group were used as phenotype label. Gene sets were acquired from previous published work[51]. Number of permutations = 100. Collapse dataset to gene symbols=FALSE. Permutation Type=phenotype. NES and FDR were used to determine statistical significance.

**Immune cell infiltration fraction analysis**. Immune cell infiltration fraction analysis was performed using the ssGSEA method based on the LM-22 immune cell marker signature[27]. Transcriptomic expression profiles were used as the input for analysis.

**TIP analysis**. Tracking Tumor Immunophenotype (TIP) analysis was performed using the website tool (http://biocc.hrbmu.edu.cn/TIP/index.jsp)[52]. Transcriptomic expression profiles were used as the input for analysis.

**Immunohistochemistry**. For immunohistochemistry, tissue sections were fixed in paraformaldehyde (4%), embedded in paraffin, and cut into 4 μm sections. Streptavidin-biotin immunostaining method was employed[28].

**Construction of mouse spontaneous glioma cell using sleeping beauty transposon**. Construction of spontaneous glioma cells (mGSC) was done using sleeping beauty transposon[53]. In brief, PEI/DNA complexes were prepared according to nucleic acid delivery protocol : vivo-jetPEI = 0.08 μl, total of DNA = 0.5 μg (PT2/C-Luc//PGK-SB13 = 0.1 μg, pT/Nestin-SV40-LgT =0.2 μg, pT/Nestin-NRASV12 = 0.2 μg). Two microliters of PEI/DNA complexes (0.5 μg/μl) were administered to neonatal mouse brain. One month later, mGSC was cultured in DMEM/F12 (Gibco), supplemented with 2% B27 supplement (Gibco), 20 ng/mL EGF (Peprotech, Rocky Hill, NJ, USA), 20 ng/mL basic-FGF (Peprotech), and 1% penicillin/streptomycin (Gibco) at 37 °C with 5% $CO_2$.

**Cell lines and cell culture**. The human breast cancer cell lines MDA-MB-231 and MDA-MB-468 were purchased from the Chinese Academy of Sciences cell bank (Shanghai, China). The human breast cancer cell lines T-47D, HCC38, human melanoma cell lines A-375 and WM-115 were purchased from iCell (Shanghai, China). Human melanoma cell line MeWo was purchased from FENGHUISHENGWU (Hunan, China). Human melanoma cell line SK-MEL-3 was purchased from COBIER (Jiangsu, China). The human monocyte cell line THP-1 was purchased from the Chinese Academy of Sciences cell bank (Shanghai, China). The mouse breast cancer cell line 4T1 and mouse melanoma cell line B16-F10 were purchased from Procell Life Science&Technology Co.,Ltd (Wuhan, China). The human breast cancer cell lines T-47D, MDA-MB-468 and human melanoma cell line A-375 were maintained in Dulbecco's Modified Eagle's Medium (DMEM, Gibco, USA) containing 10% fetal bovine serum (FBS) and 1% penicillin/streptomycin at 37 °C with 5% $CO_2$. The human breast cancer cell line MDA-MB-231 was maintained in Leibovitz's L-15 medium containing 10% FBS and 1% penicillin/streptomycin at 37 °C without $CO_2$. The human breast cancer cell line HCC38 was cultured in RPMI-1640 medium (Gibco, USA) containing 10% FBS and 1% penicillin/streptomycin (Gibco) at 37 °C with 5% $CO_2$. The human monocyte cell line THP-1 was cultured in RPMI-1640 medium (Gibco, USA) containing 10% FBS and 1% penicillin/streptomycin (Gibco) at 37 °C with 5% $CO_2$. Human melanoma cell line MeWo and WM-115 were cultured in MEM medium (Gibco, USA) containing 10% FBS and 1% penicillin/streptomycin (Gibco) at 37 °C with 5% $CO_2$. Human melanoma cell line SK-MEL-3 was cultured in McCoy's 5a medium (Gibco, USA) containing 15% FBS and 1% penicillin/streptomycin (Gibco) at 37 °C with 5% $CO_2$. The mouse breast cancer cell line 4T1 was cultured in RPMI-1640 medium (Gibco, USA) containing 10% FBS and 1% penicillin/streptomycin (Gibco) at 37 °C with 5% $CO_2$. Mouse melanoma cell line B16-F10 was cultured in RPMI-1640 medium (Gibco, USA) containing 10% FBS and 1% penicillin/streptomycin (Gibco) at 37 °C with 5% $CO_2$.

**Cell treatment**. For AKT inhibition, the cells were pre-incubated with 10 μM MK2206 (MCE, USA) for 48 h before cell co-culture were performed.

**Cell migration assay**. 8 μm transwell chamber was used for cell migration in vitro. After co-culture with tumor cells, THP-1 cells were re-suspended as single cells in RPMI-1640 medium containing 0.2% FBS and seeded into the upper chambers at a density of $4 \times 10^4$ cells/200 μL. 800 μL RPMI-1640 medium containing 20% FBS was added into the lower chamber. After 24 h of incubation, cells on the upper side of the membrane were removed using cotton swab. Cells that migrated to the lower side of the membrane were stained with 1% crystal violet solution after methanol fixation.

**Protein extraction and western blotting**. Total proteins were extracted using whole cell lysis buffer (Beyotime Biotechnology, Beijing, China) and quantified using bicinchoninic acid (BCA) method[54]. 20 μg protein was loaded, electrophoresed using 10% SDS-PAGE and transferred to a polyvinylidene difluoride (PVDF) membrane (0.45 μm; Millipore, Burlington, MA, USA). After blocking with skimmed milk (5%), the PVDF membranes were incubated with the primary antibodies overnight at 4 °C. Then, the PVDF membranes were incubated with the secondary antibodies at 25 °C for 1 h. Chemiluminescence ECL reagent (Tanon, Woburn, MA, USA) was used for protein visualization.

**Intracranial xenograft transplantation**. Male C57BL/6 N mice (6–8 weeks of age) were purchased from Beijing Vital River

Laboratory Animal Technology. Mice were raised in laminar flow cabinets under specific pathogen-free (SPF) conditions. For orthotopic transplantations, 3 μL glioma cell suspension (mGSC sphere: $1 \times 10^4$) was injected into the mouse brains at a depth of 3.0 mm using stereotactic devices.

**Subcutaneous xenograft transplantation**. Male C57BL/6 N mice (6–8 weeks of age) were purchased from Beijing Vital River Laboratory Animal Technology. Approximately $1 \times 10^6$ B16-F10 cells were transplanted into the right flank of the animals in inoculation volumes of 200 μl sterile media. Female BALB/C mice (6–8 weeks of age) were purchased from Beijing Vital River Laboratory Animal Technology. Approximately $2 \times 10^6$ 4T1 cells were transplanted into the right flank of the animals in inoculation volumes of 200 μl sterile media. The tumor mass was weighted after tumor harvest.

**In vivo inhibitor experiments**. Three days after tumor xenograft transplantation, the mice were randomly divided into control, anti-ICBs, MK2206, and anti-ICBs+MK2206 groups. MK2206 was orally administered on days 6, 8, 10, 12, 14, 16, 18, and 20 at a dose of 150 μg/g body weight. DMSO was orally administered as a control for MK2206. Anti-ICBs were intraperitoneally injected on days 7, 10, and 13 at a dose of 10 μg/g body weight. ICBs including anti-PD-L1, anti-PD-1 or anti-CTLA-4 which were bought from BioXcell, USA.

**RNA isolation and reverse-transcription quantitative PCR (RT-qPCR)**. Trizol (Invitrogen, USA) was used to extract total RNA according to manufacture's protocol. Total RNA was reversely transcribed into cDNA using Prime-Script RT Master Mix Kit (TaKaRa, Japan). RT-qPCR was performed with TB Green Premix Ex Taq II (TaKaRa, Japan). The PCR primer sequences were as follows:

Ifng Forward Primer: CAGCAACAGCAAGGCGAAAAAGG
Ifng Reverse Primer: TTTCCGCTTCCTGAGGCTGGAT
Tnfa Forward Primer: GGTGCCTATGTCTCAGCCTCTT
Tnfa Reverse Primer: GCCATAGAACTGATGAGAGGGAG
Cd86 Forward Primer: ACGTATTGGAAGGAGATTACAGCT
Cd86 Reverse Primer: TCTGTCAGCGTTACTATCCCGC
Cd206 Forward Primer: GTTCACCTGGAGTGATGGTTCTC
Cd206 Reverse Primer: AGGACATGCCAGGGTCACCTTT
Arg1 Forward Primer: CATTGGCTTGCGAGACGTAGAC
Arg1 Reverse Primer: GCTGAAGGTCTCTTCCATCACC
Trem2 Forward Primer: CTACCAGTGTCAGAGTCTCCGA
Trem2 Reverse Primer: CCTCGAAACTCGATGACTCCTC
Gapdh Forward Primer: CATCACTGCCACCCAGAAGACTG
Gapdh Reverse Primer: ATGCCAGTGAGCTTCCCGTTCAG

**CyTOF analysis of immune cells**. CyTOF analysis was performed by Novogene Co., Ltd. (Beijing, China). In brief, tumor tissue was digested by DNAase and Tyrisin to make a single-cell suspension. Cells were enriched using Percoll density gradient media, and red blood cells were removed by Red Blood Cell Lysis Buffer. The types of immune cells were identified by tSNE, followed by KNN clustering. After preliminary analysis, one sample in NC group was obviously distinct from others and was removed in further analysis. For the panels in Fig. 4, the t-SNE coordinates were determined before the sample in NC group was removed. For graphing, one sample in MK2206 group was randomly removed to keep balance.

**Construction of the tumor MT resource**. The Tumor MT website was developed in JSP using a Laravel framework and was deployed on a Nginx Web server that ran under Ubuntu system.

All data in Tumor MT was stored and managed using MySQL. jQuery was used to manage the result views. Tumor MT was thoroughly tested using Google Chrome.

**Statistics and reproducibility**. Statistical analyses were conducted using Prism 7 and R 3.5.1 unless otherwise stated. Definite statistical methods of various statistical tests are described and referenced in their respective sections. Otherwise, a two-tailed $t$ test or one-way ANOVA was used, and $p < 0.05$ was considered statistically significant. FDR adjust was done using Benjamin and Hochberg's method[45].

**Reporting summary**. Further information on research design is available in the Nature Portfolio Reporting Summary linked to this article.

## Data availability

RNA-seq data of mice glioma support the findings in this manuscript are available in BioProject database at https://ngdc.cncb.ac.cn/bioproject/, with BIGD ID: PRJCA021222. RNA- seq data of human glioma support the findings in this manuscript are available in BioProject database at https://ngdc.cncb.ac.cn/bioproject/, with BIGD ID: PRJCA021271. Numerical source data for graphs and charts in this manuscript could be found in Supplementary Data 19. Uncropped and unedited blot images could be found in Supplementary Fig. 22. All other data are available from the corresponding author upon reasonable request.

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

## Acknowledgements

The authors would like to acknowledge all the members in Dr. Wu AH's laboratory for help with this study. This work was supported by the National Natural Science Foundation of China (nos. U20A20380, 81172409, 81472360, and 81872054 to A.W., no. 81872057 to P.C., no. 81902546 to W.C., no. 82103464 to Q.L. and no. 81672824 to L.C.), Liaoning Science and Technology Plan Projects (no. 2011225034 to A.W.), the Natural Science Foundation of Liaoning Province (no. 20180550063 to P.C.), the National Postdoctoral Program for Innovative Talents (no. BX20180384 to W.C.), the China Postdoctoral Science Foundation (no. 2022M713484 to S.S. and no. 2019M651169 to W.C.), and Liao Ning Revitalization Talents Program (no. XLYC1807255 to W.C.), the Liaoning Province Science and Technology Plan Project (no. 2020-BS-117 to Q.G.)

## Author contributions

S.S., X.L., Q.G., P.C., W.C. and A.W. contributed to the study conceptualization. Q.L., X.L., Z.Y. and Q.G. contributed to data curation. A.W., W.C., P.C. and L.C. contributed to funding acquisition. J.W., C.Zou., Q.G. and C.Zhu. took charge of the experiments. G.G., L.C. and P.C. took charge of the methodology. T.L., C.Zou. and Q.G. contributed to project administration. W.C. and A.W. supervised the study and took charge of the review and editing. S.S. took charge of validation. S.S. and T.L. contributed to writing the original draft. All authors contributed to the article and approved the submitted version.

## Competing interests

The authors declare no competing interests.

## Ethical statement

The experimental protocol was approved by the ethics committee of The First Hospital of China Medical University. Collection of tumor tissue and clinicopathologic information was obtained with informed consent. All ethical regulations relevant to human research participants were followed. Animal experiments were conducted in accordance with the China Medical University Animal Care and Use Committee guidelines and approved by the Institutional Review Board of the First Hospital of China Medical University. We have complied with all relevant ethical regulations for animal use.
