## [Peer Review File · Communications Biology]

Reviewers' comments:

Reviewer #1 (Remarks to the Author):

Main:

I'm concerned that the EMT subtype reported is just a reflection of lineages. From previous work, we know that some cancer types have generally lower EMT activity score than the others. The distribution of EMT subtype in Figure 2C confirms that. The survival and other analyses have little meaning in this sense as we know some cancer types have very poor outcome mainly due to effective treatment. Similarly, we know that some mutations, CNV are more prevalent in certain cancer types than the others. The reported EMT subtypes' features could well be because of that. As such, I'm not convinced the value of such subtyping.

The authors wanted to have a pan-cancer analyses however, for most part of the data analyses and validation supporting their findings, were only reported on certain disease type. The functional assay is only on one cancer type. It is therefore unclear if the authors had cherry pick the results. In addition, this makes the paper hard to follow. My suggestion is to reorganize the paper, limit the result to those truly observed in pan-cancer, or limit to disease types where most of the findings hold true.

Major:

1. Please double check the references and the text citing it. The paraphrase has to be conveying the findings of the reference accurately and not taken out of context. For example, in Introduction, the authors cited reference 3-5 to state that somatic alterations have been the primary driving mechanism of EMT. Please specify what somatic alterations else it is confusing. I believe reference 3-5 do not specifically say main EMT driver is mutation or copy number aberration, but rather overexpression of transcription factors.

The tumor microenvironment could play a role in triggering EMT in tumor cells and it is likely EMT-ed cells acquire immunosuppressive characteristics. This is different from what was written by the authors (reference 9,10).

I'm not too sure what this sentence in line 62-63 means-"Considering the heterogeneity in EMT process, high EMT sub-types need to be established to investigate whether these unfavorable survival factors are common or unique in different individuals", what are the survival factors authors referring to? why high EMT?

2, Please be clear and avoid using ambiguous description. Please reserve keywords for its intended usage.

-For example, I'm not too sure what this sentence in line 62-63 means-"Considering the heterogeneity in EMT process, high EMT sub-types need to be established to investigate whether these unfavorable survival factors are common or unique in different individuals", what are the survival factors authors referring to? why high EMT?

- line 151, "variant type". the use of variant here is confusing as the word is usually referred to mutation.

- the authors use EMT and MT abbreviation.. which is very confusing. I suggest just stick to EMT.

3. Details and descriptions are missing for Methods and Materials.

- specify data version where applicable. if data version is not available, date when data was last accessed.
- data types for the different dataset should be mentioned. most of the data do not have multi-omics profiling. Also, the data platform for each of the dataset has to be mentioned as well.
- how was the data pre-processing or normalization done? or the authors downloaded the processed data from the data sources? what kind of normalized version of data was used?
- specify R packages version, software version.
- how was baseline mutation frequency computed for each cancer type?
- how was the CNV amplification, deletion defined? by what criteria?
- how was the in-house glioma processed? what is the accession ID? is the data publicly accessible?
- Reference 18 is for GSVA package, not ssGSEA method. while ssGSEA has been evaluated by 19, that doesn't give any credibility to the mesenchymal transition activity proposed by the authors. Evaluation of mesenchymal transition activity has to be done with appropriate validation data. What is the signature/genesets use for estimating this mesenchymal transition activity? this was not specified.
- The authors say batch effect were removed. But no data to support this.
- line 314, what is "reasonable"

4. Line 152, what is the purpose of sample number normalization? i don't see the merit of doing such normalization over to just using percentage? what is the justification?

5. A good correlation with other EMT signature does not necessarily mean the proposed signature is functionally detection EMT cells. The correlation seems to all be lower than 0.8, and more than half have correlation less than 0.6. At the very least, mean, median, the correlation should be provided. The definition of mes, no-mes subtypes were from who? what were the criteria? the authors should also mention what disease it was instead just putting the accession number or the institution name. Assuming the Mes subtype is really a bona fide Mes, predicting a higher activity in some cohorts do not prove the signature is working. i'm suspecting a stromal signature, immune signature, IFNg signature, or cell cycling signature may do the job as well. How could, using GSVA, give you such a high number 5000 as score?

Minor:

- line 62, "Sub-type" should be "subtype" for consistency.
- check abbreviations, for example, what is EMTCG in line 117
- line 170, it's "Euclidean" distance. what is $\epsilon()$?
- Figure S1, a table would do a better job.

Reviewer #2 (Remarks to the Author):

This is a very interesting study that established the core EMT signature to distinguish the tumors into 4 clusters. Importantly, the study identified two different subtypes of high EMT tumors, that is, the intrinsic mutation-related EMThigh-YAP cluster (cluster III) and the AKT activation-related EMThigh-AKT cluster (cluster IV). They further revealed that cluster IV tumors are more related to tumor immunomicroenvironment. The combination of the AKT inhibitor and ICB will be an effective therapy for cluster IV tumors. In general, the study is very comprehensive and provides important information for applying the EMT signature to clinical cancer treatment. The study deserves to be published in Communication Biology after strengthening certain points. The following are comments on the study.

1. The text in some figures is too small to be read. For example, Fig S4, Fig S5, Fig S8, Fig S10, Fig S11, Fig S12, Fig S13,
2. Fig. 2D shows cluster 4 tumors have the best survival compared to other clusters. Please provide

the potential explanation since cluster 4 is EMT-high tumors.

3. Typos in the legend of Fig. S8, e.g., burderns (burdens), muatations (mutations).

4. It would be nice if the authors could use clinical samples to demonstrate Akt phosphorylation in cluster 4 tumors by immunohistochemistry since the current study did not validate the samples at the protein level.

5. Four breast cancer cell lines are used in Figure 4 to represent different clusters of EMT. The author may explain how the cell lines are clustered.

6. The cell line experiment part is relatively weak compared to the very comprehensive multiomic analysis. For Fig. 4E-H, the study only used one cell line (MDA-MB231) to demonstrate the interaction between cancer cells and macrophages. An additional cell line to validate this finding will be helpful to prevent specific characteristics of the cell line.

7. In Fig. 5F-H, only one marker for M1 and one marker for M2 were applied. Analysis of additional markers to show the switch between M1 and M2 is suggested.

8. Analysis of clinical samples to support the findings of the study in Fig. 6 is important. Invigor210 is a clinical trial from metastatic urothelial carcinoma that is not classified as cluster 4 tumor in Fig. 2C. GSE78220 is a melanoma data set that also does not belong to cluster 4 tumor (Fig. 2C). The author may provide explanations about the clinical data set in EMT clustering.

Point to point response to reviewer's comments:

Referee expertise:

Referee #1: Pan-cancer dataset analysis

Referee #2: EMT in cancer biology and clinical oncology

Reviewers' comments:

Reviewer #1 (Remarks to the Author):

Main:

1. I'm concerned that the EMT subtype reported is just a reflection of lineages. From previous work, we know that some cancer types have generally lower EMT activity score than the others. The distribution of EMT subtype in Figure 2C confirms that. The survival and other analyses have little meaning in this sense as we know some cancer types have very poor outcome mainly due to effective treatment.

Response: We thank the reviewer for this constructive comment.

Since Epithelial-mesenchymal transition (EMT) is a common process during tumor progression and is closely related to residual tumor, drug resistance and immune suppression in tumor microenvironment, we want to investigate the EMT status of all epithelial originated tumors in pan-cancer scale. Therefore, we employed TCGA pan-cancer datasets (<https://xenabrowser.net>) and included 9415 individual samples from 24 types of epithelial originated tumors for our analysis.

In the revising process, we found that there are 1864 samples in Cluster I and 498/1864 (26.7%) of them are LGG samples. Therefore, to avoid that the identified Cluster I might be a reflection of LGG lineages and sample number bias advocated by TCGA cohorts (1), we further performed normalization and supplemented the following stratified sampling analysis for our study (Page 11, Line 179):

Sampling Analysis

To make sure our analysis didn't skew by sample number variation, we sampled 10 patients of each cancer in each cluster and merged them together to eliminate the bias due to cancer type specificity. Sampling would be repeated for 100 times to remove false positive results (1).

We consider that introducing this strategy could help us remove the false positive results due to cancer lineage and limit the result to those truly observed in pan-cancer analysis.

The revised data according to sampling strategy are as follows:

1) Over-activated pathway analysis:

In the previous version, Cluster III was identified with YAP pathway over-activation and named as EMT^{high}-YAP subtype. In this modified version, after sampling analysis, no specific over-activated pathway was identified in Cluster III. Thus, Cluster III was named as EMT^{high}-NOS (Not Otherwise Specified) subtype. For Cluster IV, we consistently observed significantly upregulated AKT pathway activity after introducing new sampling strategy. We have revised the manuscript as follows (Page 28, Line 493):

Compared to Cluster I or Cluster II, all 9 pathways were over-activated in two EMT^{high} subtypes. When pathway activity was compared between two mesenchymal like subtypes, we found that AKT pathway was over-activated in Cluster IV (100% (100/100) iterations, Fig. 3D). However, we didn't find specific over-activated pathway in Cluster III (Supplementary Fig. 9). Therefore, Cluster III was thus termed as the EMT^{high}-NOS (Not Otherwise Specified) subtype, whereas Cluster IV was termed as the EMT^{high}-AKT subtype.

Revised Figure 3D:

Revised Figure 3D: AKT pathway activity was significantly upregulated in EMT^{high}-AKT subtype; *** $p < 0.001$.

Supplementary Figure 9:

Supplementary Figure 9: Identification of driving pathways

- A. Comparison of FOXO pathway activity among four EMT subtypes in TCGA cohort.
- B. Comparison of HIF pathway activity among four EMT subtypes in TCGA cohort.
- C. Comparison of MAPK pathway activity among four EMT subtypes in TCGA cohort.
- D. Comparison of NOTCH pathway activity among four EMT subtypes in TCGA cohort.
- E. Comparison of TGFB pathway activity among four EMT subtypes in TCGA cohort.
- F. Comparison of TNF pathway activity among four EMT subtypes in TCGA cohort.
- G. Comparison of WNT pathway activity among four EMT subtypes in TCGA cohort.

H. Comparison of YAP pathway activity among four EMT subtypes in TCGA cohort.

For all graphs in this figure, Student's t test was used to analyze statistical significance between 2 groups; ****p < 0.0001; ***p < 0.001; **p < 0.01; *p < 0.05; NS p > 0.05.

2) Mutation analysis:

In the previous version, key mutations in Cluster I were IDH1, CTNNB1, ATRX and CIC; key mutations in Cluster II were TP53, PIK3CA, APC, PTEN and ARID1A; key mutations in Cluster III were TP53, KMT2D, PIK3CA and FAT1; key mutations in Cluster IV were BRAF, KRAS, VHL and PBRM1. With the new sample analysis strategy, only TP53 was identified as the key mutation in Cluster III (Revised Figure 2E, mean OR = 2.20, mean confidence interval = 1.56 ~ 3.26). We have revised the manuscript accordingly (Page 29, Line 514):

In EMT^{high}-NOS subtype, TP53 was a recurrent mutation event in 86% (86/100) iterations (median OR = 2.20, 1.50~3.26). Compare to 27.52% mutation rate in EMT^{high}-AKT subtype, TP53 mutation rate is as high as 65.78% in EMT^{high}-NOS subtype (Fig. 3F).

Revised Figure 2E:

Revised Figure 2E: Summarization of biological features among four EMT clusters.

New Figure 3F:

New Figure 3F: Compared to EMT^{high}-AKT subtype, the mutation rate of TP53 is significantly higher in EMT^{high}-NOS subtype.

3) Copy number variation analysis:

In the previous version, key CNVs in Cluster I include deletion of 1p36.23, 1p13.2, 19q13.32, 19q; key CNVs in Cluster II include amplification of 13q34, 6p22.3, 10q15.1 and deletion of 17q11.2, 15q11.2, 15q15.1; key CNVs in Cluster III include amplification of 5p15.33, 3q26.2, 8q24.21, 1p34.2 and deletion

of 9q21.3, 5q11.2, 5q21.3, 4p16.3, 3p14.2, 3p26.3; key CNVs in Cluster IV includes amplification of 5q35.3 and deletion of 3p26.3. During revision with the above stratified sampling analysis strategy, we only observed key CNVs in Cluster III, including amplification of 3q26.2 (median OR = 1.73), amplification of 11q13.3 (median OR = 1.85) and deletion of 3p14.2 (median OR = 1.65) (Revised Figure 2E). We have modified the manuscript as follows (Page 29, Line 516):

Besides, recurrent deletion of 3p14.2 (in 86% (86/100) iterations, median OR = 1.65), and amplification of 3q26.2 (in 98% (98/100) iterations, median OR = 1.73) and amplification of 11q13.3 (in 97% (97/100) iterations, median OR = 1.85) are representative CNVs in EMT^{high}-NOS subtype.

4) Clinical features in two EMT^{high} subtypes:

During revision, we found that we applied a wrong normalization strategy in the previous version (previous Page 24, Line 410). To correct this mistake, we employed the normalization strategy which is also advocated by the TCGA group (1) and made the following revision (Page 27, Line 472):

Since the sample number in each clinical aspect varied drastically (for example, whereas 1102 cases of BRCA were available, only 120 cases were available for THYM), we made normalization of sample number to make sure our analysis does not skew by sample sizes as shown in Supplementary Table 13 (1). For example, in histological type part, the sample number of each cancer type was adjusted into 100 to calculate the sample number of every cancer type in each cluster. Firstly, we compared the difference of tumor composition fraction (Fig. 3A-C and Supplementary Fig. 8A and B). Cluster III was mainly composed of squamous carcinoma, including HNSC (24.3%), LUSC(20.8%), CESC (15.8%) and ESCA (13.7%). In total, these four cancer types composited 74.7% of Cluster III samples. As for Cluster IV type, top composition cancer types are all adenocarcinoma, including KIRC (13.3%), PAAD (12.4%) and THCA (10.55%). Secondly, we found that Cluster III was mainly composed of recurrent or primary samples while the portion distribution was relatively even in Cluster IV. Thirdly, we discovered that with histological grade progression, high EMT tumor samples tended to enrich in Cluster IV type (from Grade I to Grade IV: 15.4%, 18.5%, 17.7%, and 48.4%) instead of Cluster III type (from Grade I to Grade IV: 33.6%, 36.0%, 21.3%, and 9.1%). Lastly, age and gender distribution patterns were relatively similar between these two mesenchymal like subtypes. These results demonstrated there are substantial histopathological differences between Cluster III and IV with similar enhanced mesenchymal transition status.

Revised Figure 3A-C:

Revised Figure 3A-C:

A. Histological type composition analysis of EMT^{high}-NOS and EMT^{high}-AKT subtypes showing that

EMT^{high}-NOS subtype was mainly composed of squamous carcinoma, while EMT^{high}-AKT subtype was mainly composed of adenocarcinoma.

B. Sample portion composition analysis of EMT^{high}-NOS and EMT^{high}-AKT subtypes demonstrating that EMT^{high}-NOS subtype was mainly composed of recurrent or primary samples, while the portion distribution was relatively even in EMT^{high}-AKT subtype.

C. WHO grade composition analysis of EMT^{high}-NOS and EMT^{high}-AKT subtypes showing that with histological grade progression, mesenchymal transition tumor samples tended to enrich in EMT^{high}-AKT subtype.

Revised Supplementary Figure 8A and B:

Revised Supplementary Figure 8A and B:

A. Age composition (Left panel) and gender composition (Right panel) analysis of EMT^{high}-NOS subtype.

B. Age composition (Left panel) and gender composition (Right panel) analysis of EMT^{high}-AKT subtype.

5) Immune cell infiltration analysis:

In the previous version, EMT^{high}-AKT subtype had the highest level of CD8+ T cell infiltration. With introducing the above sampling analysis strategy, EMT^{high}-AKT subtype exhibited similar CD8+ T cell infiltration with that in other Clusters. We have revised the manuscript accordingly (New Figure S12D) (Page 32, Line 565):

Compared to other subtypes, the EMT^{high}-AKT subtype exhibited similar CD8+ T cell infiltration level in 97% (97/100) iterations (Supplementary Fig. 13D).

Supplementary Figure 13D:

Supplementary Figure 13D: Comparison of CD8+ T cell proportion between the four EMT subtypes in TCGA cohort; EMT^{high}-AKT subtype had similar CD8+ T cell infiltration level with other three EMT clusters in 97 iteration times; Student's t test was used to analyze statistical significance between 2 groups; ****p < 0.0001; ***p < 0.001; **p < 0.01; *p < 0.05; NS p > 0.05.

2. Similarly, we know that some mutations, CNV are more prevalent in certain cancer types than the others. The reported EMT subtypes' features could well because of that. As such, i'm not convinced the value of such subtyping.

Response: We thank the reviewer for this helpful comment. To further support our conclusions, we analyze mutation and CNV data in TCGA pan-cancer cohort with the sampling strategy described above. Relative data could be accessed from “Mutation analysis” and “Copy number variation analysis” from Main 1.

3. The authors wanted to have a pan-cancer analyses however, for most part of the data analyses and validation supporting their findings, were only reported on certain disease type. The functional assay is only on one cancer type. it is therefore unclear if the authors had cherry pick the results. In addition, this makes the paper hard to follow. My suggestion is to reorganize the paper, limit the result to those truly observed in pan-cancer, or limit to disease types where most of the findings hold true.

Response: We appreciated this valuable comment. In previous manuscript, considering that breast cancer had the largest sample number (n=1102) in TCGA cohort, we chose breast cancer cell lines to perform functional experiments in Figure 4. In this revision, given that melanoma is the first cancer type approved by FDA to use immune checkpoint blockade (ICB) therapy (2), we employed four melanoma cell lines to validate our findings obtained from breast cancer cell lines (MEWO EMT^{low}, A375 EMT^{mid}, SKMEL3 EMT^{high}-NOS and WM115 EMT^{high}-AKT). We acquired similar results from these cell lines. The results were listed below and added into revised manuscript as follows (Page 34, Line 606):

By using melanoma cell lines, similar tumor-TAM feedback loop was also observed in WM115 cell line (EMT^{high}-AKT subtype, Supplementary Fig. 14). These results suggested that the AKT pathway was the specific hub mediating immunosuppression in EMT^{high}-AKT subtype by regulating the crosstalk between mesenchymal transition and TAM manipulation.

Supplementary Figure 14:

Supplementary Figure 14 AKT pathway is important for mediating mesenchymal transition and immunosuppression only in EMT^{high}-AKT subtype

A. Left panel: Immunoblotting analysis of N-Cadherin and Vimentin indicated that macrophage could not promote mesenchymal transition of MEWO. Right panel: Transwell analysis indicated MEWO could not enhance infiltration ability of macrophages.

B. Left panel: Immunoblotting analysis of N-Cadherin and Vimentin indicated that macrophage could not promote mesenchymal transition of A375. Right panel: Transwell analysis indicated A375 enhanced infiltration ability of macrophages mildly and could not be inhibited by AKT inhibition.

C. Left panel: Immunoblotting analysis of N-Cadherin and Vimentin indicated that macrophage could not enhance mesenchymal transition of SKMEL3. Right panel: Transwell analysis indicated SKMEL3 enhanced infiltration ability of macrophages mildly.

D. Left panel: Immunoblotting analysis of N-Cadherin and Vimentin indicated that macrophage could enhance mesenchymal transition of WM115 which could be inhibited by AKT inhibition. Right panel: Transwell analysis indicated WM115 enhanced infiltration ability of macrophages dramatically and could be inhibited by AKT inhibition.

- E. PD-L1 expression did not upregulate in co-cultured MEWO and macrophages.
- F. PD-L1 expression did not upregulate in co-cultured A375 and macrophages.
- G. PD-L1 expression did not upregulate in co-cultured SKMEL3 and macrophages.
- H. PD-L1 expression upregulated in co-cultured WM115 and macrophages which could be inhibited by AKT inhibition.

For all assays in this figure, tumor cells were pre-treated with DMSO or MK2206 for 48 hours. n = 3 replicates; Error bars indicate SD; Student's t test was used to analyze statistical significance between 2 groups; ****p < 0.0001; ***p < 0.001; **p < 0.01; *p < 0.05; NS p > 0.05.

Additionally, as shown in main 1, we have reorganized our paper to make easy to follow by removing false positive bioinformatic results and only retaining results truly observed in pan-cancer scale.

Major:

1. Please double check the references and the text citing it. The paraphrase has to be conveying the findings of the reference accurately and not taken out of context. For example, in Introduction, the authors cited reference 3-5 to state that somatic alterations have been the primary driving mechanism of EMT. Please specify what somatic alterations else it is confusing. I believe reference 3-5 do not specifically say main EMT driver is mutation or copy number aberration, but rather overexpression of transcription factors.

Response: We appreciate this thoughtful comment. We have double checked the references and revised unproper references. The following references were included in the revised manuscript (The highlighted ref is the new ref added to revised manuscript):

Ref 1: Vasan N, Baselga J, Hyman DM. A view on drug resistance in cancer. Nature. 2019;575(7782):299-309. **PMID: 31723286.**

In this article, the authors mentioned that '**Drug resistance continues to be the principal limiting factor to achieving cures in patients with cancer**'.

Ref 2: Almenawer SA, Badhiwala JH, Alhazzani W, Greenspoon J, Farrokhyar F, Yarascavitch B, et al. Biopsy versus partial versus gross total resection in older patients with high-grade glioma: a systematic review and meta-analysis. Neuro Oncol. 2015;17(6):868-81. **PMID: 25556920.**

In this article, the authors discovered that '**an upward improvement in survival time, functional recovery, and tumor recurrence rate associated with increasing extents of safe resection**'.

Ref 3: Thierry J. Epithelial-mesenchymal transitions in tumour progression. Nature reviews Cancer. 2002;2(6):442-54. **PMID: 12189386.**

In this article, the authors mentioned that '**...Acquisition of the mesenchymal phenotype has also been associated with invasive behavior...**' and '**...By understanding the processes that trigger EMT, we might also be able to prevent it. This therapeutic strategy has the potential to block metastasis and, perhaps, also prevents cancer recurrence, because micrometastases often remain after conventional surgery, radiotherapy and/or chemotherapy...**' which indicates EMT is related with residual tumor. Also in this article, the authors mentioned that '**...A large proportion of lobular breast carcinomas and diffuse gastric carcinomas 67,68 contain E-cadherin gene mutations...**' indicating EMT could be triggered directly by mutations without transcription factor (TF) alteration.

Ref 4: Thierry J, Acloque H, Huang R, Nieto M. Epithelial-mesenchymal transitions in development and disease. Cell. 2009;139(5):871-90. **PMID: 19945376.**

In this article, there is one subtitle called '**Resistance to Chemotherapy and Immunotherapy**' which indicates EMT is related to drug resistance.

Ref 5: MA N, RY H, RA J, JP T. EMT. 2016. Cell. 2016;166(1):21-45. PMID: 27368099.

In this article, this is one subtitle called '**EMT Confers Resistance to Chemotherapy and Immunotherapy**' which indicates EMT is related to drug resistance. Also in this article, the authors mentioned '**...there is probably an EMT gradient from full, to partial, to no EMT...reflecting the intrinsic molecular heterogeneity arising from diverse driver mutation profiles...**' which indicated EMT is related with genomic mutations.

Ref 6: Mani S, Guo W, Liao M, Eaton E, Ayyanan A, Zhou A, et al. The epithelial-mesenchymal transition generates cells with properties of stem cells. Cell. 2008;133(4):704-15. PMID: 18485877.

In this article, the authors mentioned that '**...their ability to found macroscopic metastases is compromised from the outset because of their limited proliferative potential...This problem may be addressed, at least in part, by the present findings, since the EMT program that enables cancer cells to disseminate from a primary tumor also promotes their self-renewal capability...**' which indicates EMT gives cancer cells unlimited proliferative ability.

Ref 7: Wang H, Bierie B, Li A, Pathania S, Toomire K, Dimitrov S, et al. BRCA1/FANCD2/BRG1-Driven DNA Repair Stabilizes the Differentiation State of Human Mammary Epithelial Cells. Molecular cell. 2016;63(2):277-92. PMID: 27373334.

In this article, the authors found that **BRCA1/FANCD2/BRG1 is essential to maintain epithelial state of breast cancer cells while BRCA1 CNV loss helps breast cancer cells redifferentiate into mesenchymal state.**

Ref 8: P C*, J W, I W, S S, V C, Z Z, et al. FOXD1-ALDH1A3 Signaling Is a Determinant for the Self-Renewal and Tumorigenicity of Mesenchymal Glioma Stem Cells. Cancer research. 2016;76(24):7219-30. PMID: 27569208.

In this article, the authors found that FOXD1-ALDH1A3 oncogenic pathway is a determinant of mesenchymal phenotype in glioma which gives glioma more malignant behavior.

Ref 9: Su S, Liu Q, Chen J, Chen J, Chen F, He C, et al. A positive feedback loop between mesenchymal-like cancer cells and macrophages is essential to breast cancer metastasis. Cancer cell. 2014;25(5):605-20. PMID: 24823638.

In this article, there is one subtitle called '**GM-CSF-Activated Macrophages Induce EMT in Breast Cancer Cells via CCL18**'. In this section, the authors found that, MCF-7 (an epithelial type breast cancer cell line) could process EMT when cocultured with GM-CSF-activated macrophages.

Ref 10: Wu J*, Shen S*, Liu T, Ren X, Zhu C, Liang Q, et al. Chemerin enhances mesenchymal features of glioblastoma by establishing autocrine and paracrine networks in a CMKLR1-dependent manner. Oncogene. 2022;41(21):3024-36.

In our previous work, we found that glioma cell could secrete Chemerin which acts on CMKLR1 of macrophages. This Chemerin-CMKLR1 axis enhances the pro-mesenchymal effects of glioma.

Ref 11-15: These five references are cited to illustrate the origin of validation datasets.

Ref 16-17: These two references are cited to illustrate the origin of CGGA dataset.

Ref 18: Louis D, Ohgaki H, Wiestler O, Cavenee W, Burger P, Jouvet A, et al. The 2007 WHO classification of tumors of the central nervous system. Acta neuropathologica. 2007;114(2):97-109. PMID: 17618441.

This is a methodological reference which guide the classification of our in-house glioma CGGA cohort.

Ref 19: Benjamini Y, Hochberg Y. Controlling the False Discovery Rate: A Practical and Powerful

Approach to Multiple Testing. *Journal of the Royal Statistical Society Series B: Methodological*. 1995;57(1):289-300.

This is a methodological reference which is used for false discovery rate (FDR) test.

Ref 20-21: These two references are cited to illustrate that GSVA R package has the function to calculate ssGSEA score and the accuracy of using ssGSEA score to predict pathway activity.

Ref 22: Wilkerson M, Hayes D. ConsensusClusterPlus: a class discovery tool with confidence assessments and item tracking. *Bioinformatics (Oxford, England)*. 2010;26(12):1572-3. **PMID: 20427518**.

This is a methodological reference which is used for cluster of TCGA cohort patients.

Ref 23: AC B, A K, RS K, AM H, W L, W L, et al. A Comprehensive Pan-Cancer Molecular Study of Gynecologic and Breast Cancers. *Cancer cell*. 2018;33(4):690-705.e9. **PMID: 29622464**.

In this article, the authors mentioned “To make sure that our analysis does not get skewed by this variation in sample sizes, we normalized number of samples for each tumor type to 100. We then grouped samples into Gyn vs non-Gyn cancers, and again adjusted size of each population to 100. Refer to Statistical Analysis section for details on the identification of significant genes.” which emphasized the importance of sample number normalization.

Ref 24: Leek J, Johnson W, Parker H, Jaffe A, Storey J. The sva package for removing batch effects and other unwanted variation in highthroughput experiments. *Bioinformatics (Oxford, England)*. 2012;28(6):882-3. **PMID: 22257669**.

This is a methodological reference of SVA R package which is used to remove the batch effects between TCGA cohorts with other validation datasets.

Ref 25: Bailey MH, Tokheim C, Porta-Pardo E, Sengupta S, Bertrand D, Weerasinghe A, et al. Comprehensive Characterization of Cancer Driver Genes and Mutations. *Cell*. 2018;173(2):371-85 e18. **PMID: 29625053**.

In this article, the authors used 26 computational tools to catalog driver genes and mutations and they finally identified **299 driver mutations**. We downloaded the driver mutation list from their work.

Ref 26: KA H, C Y, DM W, AD C, D T, S N, et al. Multiplatform analysis of 12 cancer types reveals molecular classification within and across tissues of origin. *Cell*. 2014;158(4):929-44. **PMID: 25109877**.

In this article, the authors identified 84 driver CNV peaks and we downloaded the driver CNV list from their work.

Ref 27: Zhang C, Cheng W*, Ren X, Wang Z, Liu X, Li G, et al. Tumor Purity as an Underlying Key Factor in Glioma. Clin Cancer Res. 2017;23(20):6279-91. PMID: 28754819.

We used Estimated R package to calculate tumor purity. The detailed methods could be found in our previous work.

Ref 28-29: We calculated IFNG activity and TNFa activity according to these two references.

Ref 28: P J, S G, D P, J F, A S, X H, et al. Signatures of T cell dysfunction and exclusion predict cancer immunotherapy response. *Nature medicine*. 2018;24(10):1550-8. **PMID: 30127393**.

In this article, the authors proposed a method to predict ICB response using transcriptome data which is called TIDE algorithm. We used TIDE algorithm to predict ICB response in TCGA cohorts.

Ref 30: Subramanian A, Tamayo P, Mootha VK, Mukherjee S, Ebert BL, Gillette MA, et al. Gene set enrichment analysis: a knowledge-based approach for interpreting genome-wide expression profiles. *Proc Natl Acad Sci U S A*. 2005;102(43):15545-50. **PMID: 16199517**.

This is a methodological reference of GSEA software which is used to do gene set enrichment analysis.

Ref 31: Doucette T, Rao G, Rao A, Shen L, Aldape K, Wei J, et al. Immune heterogeneity of

glioblastoma subtypes: extrapolation from the cancer genome atlas. *Cancer Immunol Res.* 2013;1(2):112-22. **PMID: 24409449.**

We downloaded immunosuppression gene sets from this work.

Ref 32: AM N, CL L, MR G, AJ G, W F, Y X, et al. Robust enumeration of cell subsets from tissue expression profiles. *Nature methods.* 2015;12(5):453-7. **PMID: 25822800.**

We downloaded the twenty-two immune cell signatures from this work.

Ref 33: L X, C D, B P, X Z, W L, G L, et al. TIP: A Web Server for Resolving Tumor Immunophenotype Profiling. *Cancer research.* 2018;78(23):6575-80. **PMID: 30154154.**

This is a methodological reference of TIP which is used to Systematically tracking the tumor immunophenotype

Ref 34: Wiesner S, Decker S, Larson J, Ericson K, Forster C, Gallardo J, et al. De novo induction of genetically engineered brain tumors in mice using plasmid DNA. *Cancer research.* 2009;69(2):431-9. **PMID: 19147555.**

This is a methodological reference which illustrates the protocol of constructing spontaneous glioma model.

Ref 35: Guo Q*, Guan G, Cao J, Zou C, Zhu C, Cheng W, et al. Overexpression of oncostatin M receptor regulates local immune response in glioblastoma. *Journal of cellular physiology.* 2019. **PMID: 30693511.**

The detail protocol of protein extraction could be found in our previous work.

Ref 36: Vasaikar SV, Deshmukh AP, den Hollander P, Addanki S, Kuburich NA, Kudaravalli S, et al. EMTome: a resource for pan-cancer analysis of epithelial-mesenchymal transition genes and signatures. *Br J Cancer.* 2021;124(1):259-69. **PMID: 33299129.**

We downloaded 76 EMT related signatures from EMTome network.

Ref 37: Q W, B H, X H, H K, M S, L S, et al. Tumor Evolution of Glioma-Intrinsic Gene Expression Subtypes Associates with Immunological Changes in the Microenvironment. *Cancer cell.* 2017;32(1):42-56.e6. **PMID: 28697342.**

We acquired the glioma molecular subtype including mesenchymal (Mes), Classical (CL) and Proneural (PN) from this work.

Ref 38: S L, J X, R D. Molecular mechanisms of epithelial-mesenchymal transition. *Nature reviews Molecular cell biology.* 2014;15(3):178-96. **PMID: 24556840.**

In this article, the authors mentioned that ‘Transforming growth factor- β (TGF β) family proteins are potent inducers of EMT, partly through the SMAD-mediated activation of EMT transcription factor expression and the subsequent SMAD-mediated control of their transcription activities.’

Ref 39: A D, RA W. New insights into the mechanisms of epithelial-mesenchymal transition and implications for cancer. *Nature reviews Molecular cell biology.* 2019;20(2):69-84. **PMID: 30459476.**

In this article, the authors mentioned that ‘This in turn enables the activation of the PI3K–AKT, ERK–MAPK, p38 MAPK and JNK pathways, promoting cell growth and proliferation, as well as cell migration and motility via induction of EMT.’

Also in this article, the author mentioned that ‘In addition to becoming invasive and motile, cells that have undergone epithelial mesenchymal transition (EMT) also acquire resistance to several drugs and chemotherapeutic agents’ in Box1.

Ref 40: Chen X, Liu T, Wu J, Zhu C, Guan G, Zou C, et al. Molecular profiling identifies distinct subtypes across TP53 mutant tumors. *JCI Insight.* 2022;7(23). **PMID: 36256461.**

In this article, the authors mentioned “Tumor protein 53 (TP53) acts as a tumor-suppressor by inducing

cell cycle arrest, cellular senescence, DNA repair, apoptosis, and changes in metabolism” which indicates TP53 mutation leads to unlimited cell proliferation.

Ref 41: Shi ZZ, Shang L, Jiang YY, Hao JJ, Zhang Y, Zhang TT, et al. Consistent and differential genetic aberrations between esophageal dysplasia and squamous cell carcinoma detected by array comparative genomic hybridization. *Clin Cancer Res.* 2013;19(21):5867-78. **PMID: 24009147.**

Amplification of 11q13.3 leads to Cyclin D1 overexpression which promotes cell proliferation.

Ref 42: Yang W, Soares J, Greninger P, Edelman EJ, Lightfoot H, Forbes S, et al. Genomics of Drug Sensitivity in Cancer (GDSC): a resource for therapeutic biomarker discovery in cancer cells. *Nucleic Acids Res.* 2013;41(Database issue):D955-61. **PMID: 23180760.**

This is a methodological reference that we acquired cancer cell line drug sensitivity data from this database.

Ref 43: Jiang Y, Zhan H. Communication between EMT and PD-L1 signaling: New insights into tumor immune evasion. *Cancer letters.* 2020; 468:72-81. **PMID: 31605776.**

In this article, the authors mentioned ‘epithelial-mesenchymal transition (EMT) plays a pivotal role in tumor immunosuppression and immune evasion. Previous studies revealed that EMT is associated with activation of different immune checkpoint molecules, including PD-L1. EMT-induced immune escape promotes cancer progression and may also provide a platform for discovery of novel therapeutic approaches and predictive biomarkers for checkpoint inhibitor therapeutic response.’ In the Abstract part.

Ref 44: Hodi FS, Wolchok JD, Schadendorf D, Larkin J, Long GV, Qian X, et al. TMB and Inflammatory Gene Expression Associated with Clinical Outcomes following Immunotherapy in Advanced Melanoma. *Cancer Immunol Res.* 2021;9(10):1202-13. **PMID: 34389558.**

In this article, the authors motioned “Treatment for patients with melanoma has been transformed in the past 10 years by the FDA approval of immuno-oncology (I-O) therapies targeting cytotoxic T lymphocyte antigen-4 (CTLA-4) and programmed death-1 (PD-1).” which indicates melanoma is the first cancer type approved by FDA with ICB therapy.

Ref 45: Mariathasan S, Turley SJ, Nickles D, Castiglioni A, Yuen K, Wang Y, et al. TGFbeta attenuates tumour response to PD-L1 blockade by contributing to exclusion of T cells. *Nature.* 2018;554(7693):544-8. **PMID: 29443960.**

We used the data of a large phase 2 trial (IMvigor210) and cited this article which is mostly related to this trial.

Ref 46: Hugo W, Zaretsky J, Sun L, Song C, Moreno B, Hu-Lieskovan S, et al. Genomic and Transcriptomic Features of Response to Anti-PD-1 Therapy in Metastatic Melanoma. *Cell.* 2016;165(1):35-44. **PMID: 26997480.**

This is a methodological reference since we used GSE78220 dataset.

Ref 47: Mak M, Tong P, Diao L, Cardnell R, Gibbons D, William W, et al. A Patient-Derived, Pan-Cancer EMT Signature Identifies Global Molecular Alterations and Immune Target Enrichment Following Epithelial-to-Mesenchymal Transition. *Clinical cancer research: an official journal of the American Association for Cancer Research.* 2016;22(3):609-20. **PMID: 26420858.**

This a pan-cancer study of EMT. In this study, the author used four markers (CDH1 (epithelial marker, E type), CDH2 (mesenchymal marker, M type), VIM (M type), and FN1 (M type)) to classify 1,934 tumors of 11 cancer types in TCGA data into E type and M type.

Ref 48: Gibbons D, Creighton C. Pan-cancer survey of epithelial-mesenchymal transition markers across the Cancer Genome Atlas. *Developmental dynamics: an official publication of the American Association of Anatomists.* 2018;247(3):555-64. **PMID: 28073171.**

In this study, the authors established a 16-gene signature to study EMT status in TCGA pan-cancer cohort.

Ref 49: Ye X, Weinberg R. Epithelial-Mesenchymal Plasticity: A Central Regulator of Cancer Progression. *Trends in cell biology*. 2015;25(11):675-86. **PMID: 26437589**.

In this review, the authors mentioned that ‘this statement of widespread association of EMT with various cancers needs to be qualified, as many clinical pathologists dispute the existence of this program and its role in generating high-grade carcinomas’ which indicates that the true role of EMT is still being questioned and we believe this is due to the lack of further investigation of the heterogeneity of EMT.

Ref 50: Marks DK, Gartrell RD, El Asmar M, Boboila S, Hart T, Lu Y, et al. Akt Inhibition Is Associated With Favorable Immune Profile Changes Within the Tumor Microenvironment of Hormone Receptor Positive, HER2 Negative Breast Cancer. *Front Oncol*. 2020; 10: 968. **PMID: 32612958**.

In this article, the authors found that after AKT inhibition with MK2206, CD3⁺CD8⁺ density was increased while expression of myeloid genes was decreased which is consistent with of CyTOF results and provide rationale for combining Akt inhibition with immunotherapy.

Ref 51: Kadiyala P, Carney S, Gauss J, Garcia-Fabiani M, Haase S, Alghamri M, et al. Inhibition of 2-Hydroxyglutarate Elicits Metabolic-reprogramming and Mutant IDH1 Glioma Immunity in Mice. *The Journal of clinical investigation*. 2020. **PMID: 33332283**.

In this article, the authors found that anti-PD-L1 could significantly enhance T memory number in IDH-mut glioma model.

Ref 52: Zhang J, Song Z, Wang H, Lang L, Yang Y, Xiao W, et al. A novel model of controlling PD-L1 expression in ALK anaplastic large cell lymphoma revealed by CRISPR screening. *Blood*. 2019;134(2):171-85. **PMID: 31151983**.

In this article, the authors mentioned that ‘Anaplastic lymphoma kinase (ALK)-positive anaplastic large-cell lymphoma (ALK⁺ ALCL) expresses a high level of PD-L1 as a result of the constitutive activation of multiple oncogenic signaling pathways downstream of ALK activity, making it an excellent model in which to define the signaling processes responsible for PD-L1 upregulation in tumor cells.’ A

Also in this article, the authors found that ALK⁺ ALCL could upregulate PD-L1 expression through IRF4 and BATF3 without the stimulation of T cells.

Ref 53: Cerezo M, Guemiri R, Druillennec S, Girault I, Malka-Mahieu H, Shen S, et al. Translational control of tumor immune escape via the eIF4F-STAT1-PD-L1 axis in melanoma. *Nat Med*. 2018;24(12):1877-86. **PMID: 30374200**.

In this article, the authors mentioned that ‘1) The rationale for the use of anti-PD-1 or anti-PD-L1 antibodies is that interferon- γ (IFN- γ) secreted by metastases-infiltrating lymphocytes will eventually lead to tumor evasion by upregulating negative immune checkpoints such as PD-L1; 2) In the absence of the T cell environment, however, tumor cell-intrinsic PD-L1 expression (may be constitutively expressed due to genetic PD-L1 locus amplification) is not typically associated with the response to immunotherapy. This explains why PD-L1 expression alone is not a strong predictive biological marker of PD-1 blockade efficacy.’ which indicates anti-PD-L1 is only effective in reactive high PD-L1 expression.

Ref 54: Y Z, J Y, D X, X M G, Z Z, J L H, et al. Disruption of tumour-associated macrophage trafficking by the osteopontin-induced colony-stimulating factor-1 signalling sensitises hepatocellular carcinoma to anti-PD-L1 blockade. *Gut*. 2019;68(9):1653-66. **PMID: 30902885**.

In this article, the authors found that osteopontin (OPN) expressed by tumor cells could chemotaxis macrophages into tumor microenvironment and macrophages could enhance the express of PD-L1 on

tumor cells through CSF1-CSF1R axis without the participation of T cells. Anti-PD-L1 antibodies is ineffective in treating liver cancer without blocking this tumor cell-macrophage interaction.

1.1 The tumor microenvironment could play a role in triggering EMT in tumor cells and it is likely EMT-ed cells acquire immunosuppressive characteristics. this is different from what was written by the authors (reference 9,10).

Response: We thank the reviewer for this helpful comment. We apologized for the description making you confused. We have revised the manuscript accordingly (Page 5, line 65):

The interplay between tumor cells and non-tumor cells facilitates the EMT process of tumor cells and EMT-ed tumor cells shape tumor microenvironment into immunosuppressive subsequently.

1.2 I'm not too sure what this sentence in line 62-63 means-"Considering the heterogeneity in EMT process, high EMT sub-types need to be established to investigate whether these unfavorable survival factors are common or unique in different individuals", what are the survival factors authors referring to? why high EMT?

Response: We apologized for these sentence writing. It is widely accepted that EMT is generally a prognosis factor associated with unfavorable survival (3), which indicates tumor cells acquiring more malignant behaviors like invasive abilities, unlimited proliferation, and drug resistance. In present study, we sought to further characterize the association of EMT and cancer patients' survival in pan-cancer scale with EMT subtyping. Indeed, we found that the survival of patients with EMT^{high}-NOS tumor is less than that of patients with EMT^{high}-AKT. To provide a clearer description of our findings, we have edited our manuscript as follows (Page 5, Line 67):

Considering the heterogeneity of the driving force (genomic alterations or TME factors) in triggering EMT process, high EMT subtypes need to be established to investigate whether EMT-ed tumors process homogeneous or heterogenous biological characteristics, for example, overall survival time.

2, Please be clear and avoid using ambiguous description. Please reserve keywords for its intended usage. -For example, I'm not too sure what this sentence in line 62-63 means-"Considering the heterogeneity in EMT process, high EMT sub-types need to be established to investigate whether these unfavorable survival factors are common or unique in different individuals", what are the survival factors authors referring to? why high EMT?

Response: We appreciate this thoughtful comment. We have thoroughly edited our manuscript to make it clear as far as possible.

Since EMT-ed is widely accepted as a poor prognostic factor(3), we sought to investigate whether all EMT-ed subtypes have worse prognosis or not. We used EMT score as a survival factor to evaluating the association of EMT-ed tumor with patients' survival. High EMT refers to groups with high EMTCG signature (defined by our work) score calculated by ssGSEA method using GSVA R package which is EMT^{high}-NOS subtype and EMT^{high}-AKT subtype in our group.

We have edited or manuscript as follows (Page 5, Line 67):

Considering the heterogeneity of the driving force (genomic alterations or TME factors) in triggering EMT process, high EMT subtypes need to be established to investigate whether EMT-ed tumors process homogeneous or heterogenous biological characteristics, for example, overall survival time.

2.1- line 151, "variant type". the use of variant here is confusing as the word is usually referred to

mutation. 沈帅

Response: We appreciate this thoughtful comment. We have used “categorical variable” to replace “variant type” in revised manuscript to make the description clearer (Page 11, Line 176):

The number of samples for each categorical variable (histological type, sample portion, WHO grade, age and gender) were normalized into 100.

2.2- the authors use EMT and MT abbreviation. which is very confusing. I suggest just stick to EMT.

Response: We appreciate this thoughtful comment. We have replaced all “MT” to “EMT”.

3. Details and descriptions are missing for Methods and Materials.

3.1- specify data version where applicable. if data version is not available, date when data was last accessed.

Response: We appreciate this constructive suggestion. We have supplemented the following information of data version or the last accessed data in the Methods and Materials section (Page 7, Line 89):

TCGA multi-omics data (clinical information ‘2018-09-13 version’, copy number variation profile ‘2016-08-16 version’, gene mutation profile ‘2016-12-29 version’, DNA methylation profile ‘Methylation450K 2016-12-29 version’, transcriptomic expression profile ‘2016-12-29 version, RNA seq’, and reverse-phase protein array profile ‘2016-08-16 version’) were downloaded from the Xena Website (<https://xenabrowser.net/>). The cancer cell line transcriptomic expression profile ‘20180929 version, RNA seq’ was downloaded from Cancer Cell Line Encyclopedia (CCLE) Website (<https://portals.broadinstitute.org/ccle/>). Colorectal cancer datasets (KFSYSCC cohort, FRENCH cohort, GSE2109 cohort, GSE37892 cohort, GSE35896 cohort, GSE23878 cohort, GSE20916 cohort, GSE17536 cohort, and GSE13067 cohort; Transcriptome) were downloaded as the authors indicated (4). Breast cancer dataset (FUSCCTNBC cohort; Transcriptome) was downloaded as the authors indicated (5). Gastric cancer datasets (ACRG cohort, KUCM cohort, KUGH cohort, MDACC cohort and SMC cohort; Transcriptome) were downloaded as the author indicated (6). IMvigor210CoreBiologies data (RNA seq) were downloaded using the R package (version 1.0.0) provided by the following website (<http://research-pub.gene.com/IMvigor210CoreBiologies/packageVersions/>). GSE78220 (RNA seq) data was downloaded from GEO website; Anti-CTLA4 clinical trial data (RNA seq) was acquired from dbGaP: <https://www.ncbi.nlm.nih.gov/geo/query/acc.cgi?acc=GSE78220> phs000452 (https://github.com/vanallenlab/VanAllen_CTLA4_Science_RNASeq_TPM/commit/3d1793629716cc1fd8e7334ea3bf593a20e6fe07) (7) and SRA: SRP067586 cohort (8).

3.2- data types for the different dataset should be mentioned. most of the data do not have multi-omics profiling. Also, the data platform for each of the dataset has to be mentioned as well.

Response: We appreciate this constructive suggestion. We have added the following information of data types of datasets we employed for our analysis to the Methods and Materials section (Page 7, Line 95):

Colorectal cancer datasets (KFSYSCC cohort, FRENCH cohort, GSE2109 cohort, GSE37892 cohort, GSE35896 cohort, GSE23878 cohort, GSE20916 cohort, GSE17536 cohort, and GSE13067 cohort; Transcriptome) were downloaded as the authors indicated (4). Breast cancer dataset (FUSCCTNBC cohort; Transcriptome) was downloaded as the authors indicated (5). Gastric cancer datasets (ACRG cohort, KUCM cohort, KUGH cohort, MDACC cohort and SMC cohort; Transcriptome) were

downloaded as the author indicated (6). IMvigor210CoreBiologies data (RNA seq) were downloaded using the R package (version 1.0.0) provided by the following website (<http://research-pub.gene.com/IMvigor210CoreBiologies/packageVersions/>). GSE78220 (RNA seq) data was downloaded from GEO website; Anti-CTLA4 clinical trial data (RNA seq) was acquired from dbGaP: phs000452

(https://github.com/vanallenlab/VanAllen_CTLA4_Science_RNASeq_TPM/commit/3d1793629716cc1fd8e7334ea3bf593a20e6fe07) (7) and SRA: SRP067586 cohort (8).

3.3- how was the data pre-processing or normalization done? or the authors downloaded the processed data from the data sources? what kind of normalized version of data was used?

Response: We appreciate this thoughtful suggestion. We downloaded processed data from the data sources. We have clarified this in the revised Methods and Materials part (Page 7, Line 106) and included the information of data source in the revised Supplementary Table S1:

We used pre-processed data provided by these authors. The details of the sample information included in current manuscript are summarized in Supplementary Table 1.

3.4- specify R packages version, software version.

Response: We appreciate this thoughtful suggestion. We have added the following information of R packages version and software version to our revised manuscript:

Page 10, Line 143: R software ‘version 3.5.1’ was used to analyze the Pearson correlation between the expression of EMTCGs and DNA methylation level (β -value).

Page 10, Line 157: Mesenchymal transition activity of individual samples was evaluated using the ssGSEA method supported by GSVA R package ‘version 1.28.0’.

Page 11, Line 164: Univariate Cox regression analysis was chosen to assess the hazard ratio of mesenchymal transition. “coxph” function in “survival” R package ‘version 3.1-11’ was used to do univariate cox regression analysis.

Page 11, Line 170: The ConsensusClusterPlus R package ‘version 1.46.0’ was used for K-Means cluster analysis of the TCGA cohort.

Page 12, Line 184: KEGG analysis was performed using the ClueGo plug-in unit ‘version 2.5.3’ provided by Cytoscape.

Page 12, Line 189: The ComBat function of the SVA R package ‘version 3.30.1’ was used to remove the batch effect between different data cohorts.

Page 13, Line 208: The Maftools R package ‘version 1.8.10’ was used to determine driver mutations in each EMT subtype.

Page 14, Line 217: Stromal and immune scores were calculated using the ESTIMATE R package ‘version 1.0.13’ (<https://sourceforge.net/projects/estimateproject/>), and tumor purity was estimated according to the formula previously described in our previously published work.

Page 14, Line 225: Gene Set Enrichment Analysis (GSEA, <http://www.broadinstitute.org/gsea/index.jsp>, version 4.3.2) was performed to explore whether the identified sets of genes showed statistical differences between the two groups using default parameters.

Page 20, Line 347: Statistical analyses were conducted using Prism 7 and R 3.5.1 unless otherwise stated.

3.5- how was baseline mutation frequency computed for each cancer type?

Response: We apologize for ignoring to provide this information. Since mutation frequencies are

varied with cancer types (For example, glioma is a low mutation cancer, in contrast to UCEC with high mutation), we compared EMTCG genes mutation rate with whole genome mutation rate in each cancer type. Baseline mutation frequency is defined as the mutation rate of the whole genome in a certain cancer type by calculating the ratio between positive mutation event number and total observed event number. We have made the following revision to our manuscript (Page 9, Line 137):

Baseline mutation frequency is defined as the mutation rate of the whole genome in a certain cancer type by calculating the ratio between positive mutation event number and total observed event number.

3.6- how was the CNV amplification, deletion defined? by what criteria?

Response: We thank the reviewer for this comment. We have added this information to our revised manuscript (Page 10, Line 150):

Numeric focal-level Copy Number Variation (CNV) values were acquired by the Xena Website using TCGA FIREHOSE pipeline based on GISTIC2.

3.7- how was the in-house glioma processed? what is the accession ID? is the data publicly accessible?

Response: The corresponding author, Professor Anhua Wu, is a principal member of Chinese Glioma Cooperative Group (CGCG) (9, 10). Therefore, we employed Chinese Glioma Glioma Atlas (CGGA) as in-house glioma dataset. The CGGA glioma data (n=1018) is **publicly accessible** which could be downloaded at CGGA website (<http://www.cgga.org.cn>). We have modified the materials and methods part as follows (Page 10, Line 150):

A total of 1018 glioma samples were included in our study (9, 10). The details of the sample information included in this work are summarized in Supplementary Table 1. The CGGA cohort data could be obtained from the CGGA database (<http://www.cgga.org.cn>).

Besides, in this modified version, we collected another cohort of glioma patients from our institution. By using RNA-seq technique, we acquired the transcriptome of our cohort glioma tissues. We have added this information to Materials and Methods part (Page 8, Line 118):

CMU1h glioma samples

A total of 292 multiregional glioma tissue samples from 71 patients were acquired at the First Affiliated Hospital of China Medical University (CMU1H) with the assistance of neuronavigation. Samples were sent for RNA seq and classified into four EMT subtypes according to transcriptomic profile. 20 samples with sufficient tissue block were sent for immunohistochemistry staining. The sample information of CMU1H could be found in Supplementary Table 1.

3.8- Reference 18 is for GSVA package, not ssGSEA method. while ssGSEA has been evaluated by 19, that doesn't give any credibility to the mesenchymal transition activity proposed by the authors. Evaluation of mesenchymal transition activity has to be done with appropriate validation data. What is the signature/genesets use for estimating this mesenchymal transition activity? this was not specified.

Response: Thanks for this thoughtful comment. We used GSVA R package (version 1.28.0) to calculate ssGSEA score by using “method = c("ssgsea")” parameter. In the software manual of GSVA, the authors stated ssGSEA had been integrated into GSVA R package. To demonstrate the validity of our signature, we have calculated the correlation between our EMT score with other gene signatures which shows significant correlation (mean correlation co-efficiency = 0.71, Std = 0.14) with our signatures. The data was shown in Figure S2A and S2B. We have revised the manuscript accordingly (Page 22, Line

374):

To test our signature's accuracy, we downloaded other 76 EMT related signatures from EMTome (11) website (<http://www.emtome.org>). We calculated the correlation between our signature and these 76 signatures based on ssGSEA score. We found out that our signature was strongly positively correlated with these 76 signatures (mean correlation=0.71, Std=0.14, Supplementary Fig. 1A). Besides, we also calculated the correlation among each two signatures of 76 signatures and we found that the correlation (mean correlation=0.71, Std=0.14) between our signature and other 76 signatures is better than the correlation (mean correlation=0.63, Std=0.20) among each two signatures of 76 signatures (Supplementary Fig. 1B and C).

Supplementary Figure 1 A, New Supplementary Figure 1B and C:

Supplementary Figure 1 A, New Supplementary Figure 1B and C:

A. EMTCG signature shown highly positive correlation with other EMT related signatures.

B. Each EMT related signature shown positive correlation with other EMT related signatures.

C. The correlation (mean correlation co-efficiency = 0.71, Std = 0.14) between EMTCG signature with other 76 signatures is stronger than the correlation (mean correlation co-efficiency = 0.63, Std = 0.20) among each two signatures of other 76 already published signatures; Group A: correlation among each two signatures of 76 other published signatures; Group B: correlation between our EMTCG signature with other 76 published signatures.

3.9- The authors say batch effect were removed. But no data to support this.

Response: As shown in Supplementary Figure 10A, the PCA analysis indicates that TCGA cohort and other validation cohorts are distinct cohorts which may bring batch bias into analysis. Thus, we used “sva R” package to remove batch effect. The PCA analysis after “sva R” package performance indicates that batch effect was removed.

Revised Supplementary Figure 10A:

Revised Supplementary Figure 10A: PCA analysis demonstrating that External cohort and TCGA cohort are eccentric before removing batch effect and show concentric after removing batch effect.

3.10- line 314, what is "reasonable"

Response: We have revised our manuscript according to this helpful comment (Page 22, Line 368):

Then, we conducted protein-protein interaction analysis using STRING website and identified 58 genes exhibiting strong protein interactions (Supplementary Table 4).

The method to identify strong protein interaction could be found in the Materials and Methods section (Page 9, Line 128):

PPI was performed using String Website (<https://string-db.org/>). Gene ID was used as input data. The interaction network using combined score was then imported into Cytoscape 'version 3.6.1' (<https://cytoscape.org>) for network analysis. The genes that had a degree (K) within the top two-thirds of all genes were considered node genes.

4. Line 152, what is the purpose of sample number normalization? i don't see the merit of doing such normalization over to just using percentage? what is the justification?

Response: Thanks for this comment. We performed sample number normalization to avoid bias caused by sample number difference. For example, there are 1102 samples of breast cancer in TCGA cohort while there are only 120 samples of thymoma in TCGA cohort. The proportion of thymoma would be reduced. Before sample number normalization, the proportion of thymoma in each cluster is 1.3%, 2.3%, 0.1% and 0.8%. After sample number normalization, the proportion of thymoma in each cluster is 4.19%, 7.17%, 0.22% and 2.77%. This normalization method has been validated by TCGA cohorts (1).

5. A good correlation with other EMT signature does not necessarily mean the proposed signature is functionally detection EMT cells. The correlation seems to all be lower than 0.8, and more than half have correlation less than 0.6. At the very least, mean, median, the correlation should be provided.

Response: We thank the reviewer for this constructive suggestion. Firstly, our signature is established by calculating the subset of EMT related Gene Ontology (GO) gene sets through protein-protein-interaction and gene co-expression analysis to choose proper core genes in EMT analysis. The correlation of our EMTCG sets show strong correlation with other 76 EMT gene signatures downloaded from EMTome (11) (mean correlation co-efficiency = 0.71, Std = 0.14). Besides, as shown in Supplementary Figure 2B and C, the correlation between our signature with other 76 signatures (mean correlation co-efficiency = 0.71, Std = 0.14) is better than the correlation among each two signatures of other 76

published signatures (mean correlation co-efficiency = 0.63, Std = 0.20).

Supplementary Figure 1A-C could be found in major 3.8 part of reviewer 1 above.

5.1 The definition of mes, non-mes subtypes were from who? what were the criteria?

Response: The mes and non-mes subtype are defined by the authors providing these external validation cohorts employed in present work, including colorectal cancer (KFSYSCC cohort, FRENCH cohort, GSE2109 cohort, GSE37892 cohort, GSE35896 cohort, GSE23878 cohort, GSE20916 cohort, GSE17536 cohort, and GSE13067 cohort), breast cancer (FUSCCTNBC cohort), and gastric cancer cohort (ACRG cohort, KUCM cohort, KUGH cohort, MDACC cohort and SMC cohort).

5.2 the authors should also mention what disease it was instead just putting the accession number or the institution name. Assuming the Mes subtype is really a bona fide Mes, predicting a higher activity in some cohorts do not prove the signature is working. i'm suspecting a stromal signature, immune signature, IFNg signature, or cell cycling signature may do the job as well. How could, using GSVA, give you such a high number 5000 as score?

Response: We thank the reviewer for this thoughtful comment. As mentioned above (major 3.8 and major 5 of reviewer 1), our signature is established by calculating the subset of EMT related Gene Ontology (GO) gene sets through protein-protein-interaction and gene co-expression analysis to choose proper core genes in EMT analysis. Besides, the correlation of our EMTCG sets show strong correlation with other 76 EMT gene signatures downloaded from EMTome (11). Therefore, we conclude that our signature could work to identify patients with high mesenchymal status.

We used ssGSEA function in GSVA R package to estimate pathway activity and therefore we could acquire scores as high as 5000 since we didn't choose Norm parameter (Norm=F) in the GSVA R package and we used raw original result of ssGSEA for further analysis.

Minor:

1- line 62, "Sub-type" should be "subtype" for consistency.

Response: We apologized for this mistake and we have revised the manuscript accordingly (Page 5, Line 68):

High EMT subtypes need to be established to investigate whether EMT-ed tumors process homogeneous or heterogeneous biological characteristics.

2- check abbreviations, for example, what is EMTCG in line 117

Response: We apologized for this negligence. We have added the complete spelling for "EMTCG" into the manuscript (Page 10, Line 143):

R software 'version 3.5.1' was used to analyze the Pearson correlation between the expression of epithelial-mesenchymal transition core genes (EMTCGs) and DNA methylation level (β -value).

3- line 170, it's "Euclidean" distance. what is ε ()?

Response: ε () means summation. We have added this information to the revised manuscript (Page 13, Line 198):

ε means summation.

4- Figure S1, a table would do a better job.

Response: Thanks for this thoughtful comment. We have integrated Supplementary Figure 1 into revised Supplementary Table 4.

Reference:

1. AC B, A K, RS K, AM H, W L, W L, et al. A Comprehensive Pan-Cancer Molecular Study of Gynecologic and Breast Cancers. *Cancer cell*. 2018;33(4):690-705.e9.
2. Hodi FS, Wolchok JD, Schadendorf D, Larkin J, Long GV, Qian X, et al. TMB and Inflammatory Gene Expression Associated with Clinical Outcomes following Immunotherapy in Advanced Melanoma. *Cancer Immunol Res*. 2021;9(10):1202-13.
3. T S, RA W. EMT, CSCs, and drug resistance: the mechanistic link and clinical implications. *Nature reviews Clinical oncology*. 2017;14(10):611-29.
4. Guinney J, Dienstmann R, Wang X, de Reynies A, Schlicker A, Sonesson C, et al. The consensus molecular subtypes of colorectal cancer. *Nat Med*. 2015;21(11):1350-6.
5. Jiang YZ, Ma D, Suo C, Shi J, Xue M, Hu X, et al. Genomic and Transcriptomic Landscape of Triple-Negative Breast Cancers: Subtypes and Treatment Strategies. *Cancer Cell*. 2019;35(3):428-40 e5.
6. Oh SC, Sohn BH, Cheong JH, Kim SB, Lee JE, Park KC, et al. Clinical and genomic landscape of gastric cancer with a mesenchymal phenotype. *Nat Commun*. 2018;9(1):1777.
7. Van Allen E, Miao D, Schilling B, Shukla S, Blank C, Zimmer L, et al. Genomic correlates of response to CTLA-4 blockade in metastatic melanoma. *Science (New York, NY)*. 2015;350(6257):207-11.
8. Nathanson T, Ahuja A, Rubinsteyn A, Aksoy B, Hellmann M, Miao D, et al. Somatic Mutations and Neoepitope Homology in Melanomas Treated with CTLA-4 Blockade. *Cancer immunology research*. 2017;5(1):84-91.
9. Jiang T, Mao Y, Ma W, Mao Q, You Y, Yang X, et al. CGCG clinical practice guidelines for the management of adult diffuse gliomas. *Cancer Lett*. 2016;375(2):263-73.
10. Jiang T, Nam DH, Ram Z, Poon WS, Wang J, Boldbaatar D, et al. Clinical practice guidelines for the management of adult diffuse gliomas. *Cancer Lett*. 2021;499:60-72.
11. Vasaikar SV, Deshmukh AP, den Hollander P, Addanki S, Kuburich NA, Kudaravalli S, et al. EMTome: a resource for pan-cancer analysis of epithelial-mesenchymal transition genes and signatures. *Br J Cancer*. 2021;124(1):259-69.

Reviewer #2 (Remarks to the Author):

This is a very interesting study that established the core EMT signature to distinguish the tumors into 4 clusters. Importantly, the study identified two different subtypes of high EMT tumors, that is, the intrinsic mutation-related EMThigh-YAP cluster (cluster III) and the AKT activation-related EMThigh-AKT cluster (cluster IV). They further revealed that cluster IV tumors are more related to tumor immunomicroenvironment. The combination of the AKT inhibitor and ICB will be an effective therapy for cluster IV tumors. In general, the study is very comprehensive and provides important information for applying the EMT signature to clinical cancer treatment. The study deserves to be published in *Communication Biology* after strengthening certain points. The following are comments on the study.

Response: We thank the reviewer for this favorable comment.

1. The text in some figures is too small to be read. For example, Fig S4, Fig S5, Fig S8, Fig S10, Fig S11, Fig S12, Fig S13,

Response: Thanks for this thoughtful comment. We have modified the text in these figures accordingly and added the revised figure as Revised Supplementary Figure 3, Supplementary Figure 4, Supplementary Figure 7, Supplementary Figure 9, and Supplementary Figure 10.

Revised Supplementary Figure 3 (previous Supplementary Figure 4):

Revised Supplementary Figure 3:

Mesenchymal transition was a poor prognosis indicator

A. Kaplan–Meier survival analysis of overall survival of 9415 samples across 24 cancer types based on the EMT score in TCGA cohort. The red and blue curves indicate the survival curves of patients in high mesenchymal transition activity and low mesenchymal transition activity, respectively. $p < 0.05$ was considered as statistically significant.

B. Kaplan–Meier survival analysis of overall survival of each cancer types based on the EMT score in TCGA cohort. The red and blue curves indicate the survival curves of patients in high mesenchymal transition activity and low mesenchymal transition activity, respectively. $p < 0.05$ was considered as statistically significant.

Revised Supplementary Figure 4 (previous Supplementary Figure 5):

Revised Supplementary Figure 4:

Supplementary Figure 4 Pairwise comparison of the mesenchymal transition activity in TCGA cohort across 16 cancer types between tumor and non-tumor tissue.

Mesenchymal transition activity was compared in 16 cancer types in TCGA cohort with paired normal tissue and tumor tissue. Plots in the left of the graph indicated mesenchymal transition activity of paired normal tissue while plots in the right of the graph indicated mesenchymal transition activity of paired tumor tissue. These graphs indicated although the mesenchymal transition activity was enhanced in most tumor compartments, the mesenchymal transition activity was inhibited compared to paired normal samples in some tumor compartments.

Revised Supplementary Figure 7 (previous Supplementary Figure 10):

Revised Supplementary Figure 7:

Tumor Microenvironment Analysis

A. Comparison of Stromal Score among four EMT subtypes in TCGA cohort. EMT^{high}-AKT subtype had the highest Stromal Score among the four EMT subtypes in 100 iteration repeats; Student's t test was used to analyze statistical significance between 2 groups; ****p < 0.0001; ***p < 0.001; **p < 0.01; *p < 0.05.

B. Comparison of Immune Score among four EMT subtypes in TCGA cohort. EMT^{high}-AKT subtype had the highest Immune score among the four EMT subtypes in 100 iteration repeats; Student's t test was used to analyze statistical significance between 2 groups; ****p < 0.0001; ***p < 0.001; **p < 0.01; *p < 0.05.

C. Comparison of tumor purity among four EMT subtypes in TCGA cohort. EMT^{high}-AKT subtype had the lowest tumor purity among the four EMT subtypes in 100 iteration repeats; the Student's t test was used to analyze statistical significance between 2 groups; ****p < 0.0001; ***p < 0.001; **p < 0.01; *p < 0.05.

D. AKT pathway score was negatively correlated with tumor purity in EMT^{high}-AKT subtype in TCGA cohort.

Revised Supplementary Figure 9 (previous Supplementary Figure 12):

Revised Supplementary Figure 9:

Identification of driving pathways

- A. Comparison of FOXO pathway activity among four EMT subtypes in TCGA cohort.
- B. Comparison of HIF pathway activity among four EMT subtypes in TCGA cohort.
- C. Comparison of MAPK pathway activity among four EMT subtypes in TCGA cohort.
- D. Comparison of NOTCH pathway activity among four EMT subtypes in TCGA cohort.
- E. Comparison of TGFB pathway activity among four EMT subtypes in TCGA cohort.
- F. Comparison of TNF pathway activity among four EMT subtypes in TCGA cohort.
- G. Comparison of WNT pathway activity among four EMT subtypes in TCGA cohort.
- H. Comparison of YAP pathway activity among four EMT subtypes in TCGA cohort.

For all graphs in this figure, Student's t test was used to analyze statistical significance between 2 groups; ***p < 0.0001; **p < 0.001; *p < 0.01; *p < 0.05; NS p > 0.05.

Revised Supplementary Figure 10 (previous Supplementary Figure 13):

Revised Supplementary Figure 10:

AKT pathway is overactivated in EMT^{high}-AKT subtype in cancer cell lines

A. Schematic plot to reclassify individual samples into 35 EMTCGs-based EMT subtypes.

B. Comparison of AKT pathway activity score among different EMT clusters in CCLE dataset; EMT^{high}-AKT subtype had the highest AKT pathway activity among the four EMT subtypes; Student's t test was used to analyze statistical significance between 2 groups; ****p < 0.0001; ***p < 0.001; **p < 0.01; *p < 0.05.

2. Fig. 2D shows cluster 4 tumors have the best survival compared to other clusters. Please provide the potential explanation since cluster 4 is EMT-high tumors.

Response: We thank the reviewer for this comment. In the initial version, as Fig.2D shows, cluster III has the worst survival, while cluster I, II, and IV have similar survival (medial survival for Cluster I, II, III, and IV = 115.7, 101.4, 48.6, and 108.6 months, respectively). In this revised manuscript, we used the following sample analysis steps to eliminate the bias due to cancer type specificity in survival analysis: We sampled 10 patients in each cancer type, merged them into corresponding cluster, and then performed data analysis for 100 times. The results revealed that in **96% (96/100)** iterations, Cluster III still has the shortest medial survival time, while in **68% (68/100)** iterations, Cluster I have the longest medial survival time. Thus, we have made the following revision to our manuscript (Page 26, Line 448):

Besides, the survival time was also compared, showing that Cluster III had the shortest survival (median survival = 48.6 months), whereas Cluster I, Cluster II, and Cluster IV exhibited relatively long survival (medial survival for Cluster I = 115.7 months, medial survival for Cluster II = 101.4 months, medial survival for Cluster IV = 108.6 months) (Fig. 2D). To eliminate the bias due to cancer type specificity, we also introduced sampling strategy mentioned above. After sampling analysis, in 96% (96/100) iterations, Cluster III still had the shortest survival. These analysis indicated two EMT^{high} subtype might exist which urged us to investigate the heterogeneity between different EMT state tumors.

Revised Figure 2D:

Revised Figure 2D:

Kaplan-Meier curve of overall survival of patients with different EMT subtypes for TCGA datasets. Cluster III was statistically associated with unfavorable survival outcome.

3. Typos in the legend of Fig. S8, e.g., burderns (burdens), muatations (mutations).

Response: We apologized for the typos in figure legend of previous Supplementary Figure 8. In this revised version, we have removed previous Supplementary Figure 8.

4. It would be nice if the authors could use clinical samples to demonstrate Akt phosphorylation in cluster 4 tumors by immunohistochemistry since the current study did not validate the samples at the protein level.

Response: We thank the reviewer for this comment. To validate our current conclusion that EMT^{high}-AKT subtype is AKT pathway over-activated, we collected a cohort of clinical glioma samples (292 multiregional tissue sample from 71 patients) from our institution. We acquired the transcriptome data of these samples with RNA-seq technique and classified them into the corresponding subtypes proposed by current work. Then, we examined phosphorylation level of AKT protein in these samples by immunohistochemical staining and found that phosphorylation level of AKT protein was higher in EMT^{high}-AKT subtype than other three subtypes (EMT^{low}, EMT^{mid}, and EMT^{high}-NOS subtypes). We have added this part into the manuscript (Page 29, Line 499):

To further demonstrate Cluster IV is specified with AKT pathway overactivation, a cohort of glioma samples were used for IHC analysis. The result indicated the phosphorylation level of AKT pathway was significantly upregulated in Cluster IV (Supplementary Fig. 11).

New Supplementary Figure 11:

Supplementary Figure 11 Immunohistochemistry indicates phosphorylation level of AKT is significantly upregulated in EMT^{high}-AKT subtype; Scale bar: 30 μM; **p < 0.01; ***p < 0.001; ****p < 0.0001.

5. Four breast cancer cell lines are used in Figure 4 to represent different clusters of EMT. The author may explain how the cell lines are clustered.

Response: We thank the reviewer for this comment. Breast cancer cell lines employed in present study were clustered with three steps. Firstly, we removed the batch effect among individual cohorts in TCGA. Secondly, we calculate the Euclidean distance between individual samples with cluster center of the four clusters. Lastly, the individual samples are classified into the cluster with nearest distance to cluster center. The detailed method could be found in material and method part (Page 12, Line 193):

Identification of EMT Subtypes of Non-TCGA Samples

EMT subtypes of non-TCGA samples were identified with following steps: 1) The batch effect between external cohorts and TCGA cohort was removed; 2) Each individual sample in external cohort was scaled with TCGA cohort separately; 3) Euclidian distance between individual samples and four TCGA EMT cluster centers were calculated using the expression of 35 MTCG genes. Euclidian distance= $\sqrt{\varepsilon(nonTCGAi - TCGAi)^2}$, $i=35$, ε means summation; 4) Individual samples were classified into the corresponding subtype with the shortest distance.

Supplementary Figure 10A:

Supplementary Figure 10A: Schematic plot to reclassify individual samples into 35 EMTcgs-based EMT subtypes.

6. The cell line experiment part is relatively weak compared to the very comprehensive multiomic analysis. For Fig. 4E-H, the study only used one cell line (MDA-MB231) to demonstrate the interaction between cancer cells and macrophages. An additional cell line to validate this finding will be helpful to prevent specific characteristics of the cell line.

Response: We thank the reviewer for this thoughtful comment. We chose breast cancer cell lines to perform experiments in Figure 4, since breast cancer had the largest sample number ($n = 1102$) in TCGA cohort. Since melanoma is the first cancer type approved by FDA to use immune checkpoint blockade (ICB) therapy (1), we employed melanoma (SKCM) cell lines to perform functional assays in Figure 4 to further support our findings. We acquired four melanoma cell lines (MEWO EMT^{low}, A375 EMT^{mid}, SKMEL3 EMT^{high}-NOS and WM115 EMT^{high}-AKT) that could be classified into the four clusters in our work. We acquired similar results from these cell lines. The results were listed below and added into revised manuscript as follows (Page 34, Line 606):

By using melanoma cell lines, similar tumor-TAM feedback loop was also observed in WM115 cell line (EMT^{high}-AKT subtype, Supplementary Fig. 14). These results suggested that the AKT pathway was the specific hub mediating immunosuppression in EMT^{high}-AKT subtype by regulating the crosstalk between mesenchymal transition and TAM manipulation.

Supplementary Figure 14:

Supplementary Figure 14:

AKT pathway is important for mediating mesenchymal transition and immunosuppression only in EMT^{high}-AKT subtype

A. Left panel: Immunoblotting analysis of N-Cadherin and Vimentin indicated that macrophage could not promote mesenchymal transition of MEWO. Right panel: Transwell analysis indicated MEWO could not enhance infiltration ability of macrophages.

B. Left panel: Immunoblotting analysis of N-Cadherin and Vimentin indicated that macrophage could not promote mesenchymal transition of A375. Right panel: Transwell analysis indicated A375 enhanced infiltration ability of macrophages mildly and could not be inhibited by AKT inhibition.

C. Left panel: Immunoblotting analysis of N-Cadherin and Vimentin indicated that macrophage could not enhance mesenchymal transition of SKMEL3. Right panel: Transwell analysis indicated SKMEL3 enhanced infiltration ability of macrophages mildly.

D. Left panel: Immunoblotting analysis of N-Cadherin and Vimentin indicated that macrophage could enhance mesenchymal transition of WM115 which could be inhibited by AKT inhibition. Right panel: Transwell analysis indicated WM115 enhanced infiltration ability of macrophages dramatically and

could be inhibited by AKT inhibition.

E. PD-L1 expression did not upregulate in co-cultured MEWO and macrophages.

F. PD-L1 expression did not upregulate in co-cultured A375 and macrophages.

G. PD-L1 expression did not upregulate in co-cultured SKMEL3 and macrophages.

H. PD-L1 expression upregulated in co-cultured WM115 and macrophages which could be inhibited by AKT inhibition.

For all assays in this figure, tumor cells were pre-treated with DMSO or MK2206 for 48 hours. n = 3 replicates; Error bars indicate SD; Student's t test was used to analyze statistical significance between 2 groups; ****p < 0.0001; ***p < 0.001; **p < 0.01; *p < 0.05; NS p > 0.05.

7. In Fig. 5F-H, only one marker for M1 and one marker for M2 were applied. Analysis of additional markers to show the switch between M1 and M2 is suggested.

Response: We thank the reviewer for this comment. Due to the technique limitations of CyTOF analysis, only 38 antibodies could be applied in one panel. Since our main purpose is to test the toxicity effect of T cells, there are limited choice space for M1 and M2 markers. To address this comment, we perform qPCR analysis of xenograft derived from sleeping beauty transposon derived mGSC (2) to detect the status of TAMs with the indicated M1 and M2 markers to enrich the results in Figure 5F-H. The new data was added into the revised manuscript as follows (Page 35, Line 627):

To further explore the functional status of TAMs, mRNA expression level of TAM's proinflammatory (M1) or anti-inflammatory (M2) markers were also detected using RT-qPCR test. As shown in Supplementary Fig. 15, M1 markers including *Ifng* (Supplementary Fig. 15A), *Tnfa* (Supplementary Fig. 15B) and *Cd86* (Supplementary Fig. 15C) didn't alter while M2 markers including *Cd206* (Supplementary Fig. 15D), *Arg1* (Supplementary Fig. 15E) and *Trem2* (Supplementary Fig. 15F) downregulated in mGSCs derived xenograft model after MK2206 administration. These results indicated that AKT pathway inhibition could block M2 transformation of macrophages.

New Supplementary Figure 15:

New Supplementary Figure 15:

AKT pathway inhibition could block M2 transformation of macrophages in tumor tissue

- A. Ifng expression didn't alter in mGSC xenograft model after MK2206 administration.
- B. Tnfa expression didn't alter in mGSC xenograft model after MK2206 administration.
- C. Cd86 expression didn't alter in mGSC xenograft model after MK2206 administration.
- D. Cd206 expression downregulated in mGSC xenograft model after MK2206 administration.
- E. Arg1 expression downregulated in mGSC xenograft model after MK2206 administration.
- F. Trem2 expression downregulated in mGSC xenograft model after MK2206 administration.

n = 4 replicates; Error bars indicate SD; Student's t test was used to analyze statistical significance between 2 groups; ns $p > 0.05$; * $p < 0.05$.

8. Analysis of clinical samples to support the findings of the study in Fig. 6 is important. Imvigor210 is a clinical trial from metastatic urothelial carcinoma that is not classified as cluster 4 tumor in Fig. 2C. GSE78220 is a melanoma data set that also does not belong to cluster 4 tumor (Fig. 2C). The author may provide explanations about the clinical data set in EMT clustering.

Response: We thank the reviewer for this comment. As shown in Figure 6A, there are 28.5% samples in Imvigor210 cohort are classified into Cluster IV. Moreover, as shown in Fig.2C, according to urothelial carcinoma data in TCGA, 15% of BLCA, 92% of KIRC, 18% percent of KIRP, and 34% of PRAD are classified into Cluster IV, respectively. Additionally, as shown in Fig.2C, 24% patients in SKCM from TCGA are classified into Cluster IV. Collectively, there are urothelial carcinoma and melanoma samples

that could be classified into cluster IV, and there are samples in Imvigor210 and GSE78220 which could be classified as Cluster IV tumor.

References:

1. Hodi FS, Wolchok JD, Schadendorf D, Larkin J, Long GV, Qian X, et al. TMB and Inflammatory Gene Expression Associated with Clinical Outcomes following Immunotherapy in Advanced Melanoma. *Cancer Immunol Res.* 2021;9(10):1202-13.
2. Wu J, Shen S, Liu T, Ren X, Zhu C, Liang Q, et al. Chemerin enhances mesenchymal features of glioblastoma by establishing autocrine and paracrine networks in a CMKLR1-dependent manner. *Oncogene.* 2022;41(21):3024-36.

Reviewers' comments:

Reviewer #1 (Remarks to the Author):

In general, the authors have addressed most of my concerns satisfactorily.

The re-sampling analysis is ok, however, showing a particular iteration is not recommended. For example, result at 68th iterations, result at 1st iteration. Instead, i suggests
(1) the authors to put result of iterations in supplementary figure instead of putting them side by side with the main result, which distract readers.
(2) plot out the mean ssGSEA score for each cluster in all iterations with a line/trend plot instead of selectively show a certain iterations.

use proper cell line name, eg, "MDA-MB-231" instead of "MDA-231" so as to be consistent with other section of the manuscript. Molecular subtype of breast cancer cell lines should be provided as well (luminal, basal-A, claudin-low). How were the cell lines classified into EMThigh/low, AKT etc i didn't see the description or analyses anywhere in the paper.

Reviewer #2 (Remarks to the Author):

The comments of the reviewers has been adequately addressed and the revised manuscript is significant improved. The current version is acceptable for publication in Communication Biology.

Referee expertise:

Referee #1: Pan-cancer dataset analysis

Referee #2: EMT in cancer biology and clinical oncology

Reviewers' comments:

Reviewer #1 (Remarks to the Author):

In general, the authors have addressed most of my concerns satisfactorily.

Response: We thank the reviewer for this favorable comment and the new constructive suggestions to further help us to polish our work.

1. The re-sampling analysis is ok, however, showing a particular iteration is not recommended. For example, result at 68th iterations, result at 1st iteration. Instead, i suggests (1) the authors to put result of iterations in supplementary figure instead of putting them side by side with the main result, which distract readers. (2) plot out the mean ssGSEA score for each cluster in all iterations with a line/trend plot instead of selectively show a certain iteration.

Response: We appreciate this helpful comment. In the revised manuscript, we have moved the results of iterations in main figure into supplementary figures and provided the summary of 100 iteration sampling analysis by showing the mean value and data distribution of every iteration.

Revised Figure 2B:

Figure 2B: Comparison of mesenchymal transition activity among four clusters. The mesenchymal transition activity was significantly up-regulated in Cluster III and IV; the Student's t test was used to analyze statistical significance; **p < 0.01; ***p < 0.001.

Revised Figure 2D:

Figure 2D: Kaplan-Meier curve of overall survival of patients with different EMT subtypes for TCGA datasets.

Revised Figure 3D:

Figure 3D: AKT pathway activity was significantly upregulated in EMT^{high}-AKT subtype.

New Supplementary Figure 6:

Supplementary Figure 6 Two EMT^{high} subtypes have different survival status.

A. Comparison of mesenchymal transition activity among four clusters. The mesenchymal transition activity was significantly up-regulated in Cluster III and IV; the Student's t test was used to analyze statistical significance; ** $p < 0.01$; *** $p < 0.001$; left panel: Comparison in pancancer scale; right panel: Summary of 100 iterations, each point represents mean ssGSEA score in each iteration.

B. Kaplan-Meier curve of overall survival of patients with different EMT subtypes for TCGA datasets. Cluster III was statistically associated with unfavorable survival outcome; left panel: Comparison in pan-cancer scale; right panel: Summary of 100 iterations, each point represents mean survival in each iteration.

Revised Supplementary Figure 7:

Supplementary Figure 7 Tumor Microenvironment Analysis

A. Comparison of Stromal Score among four EMT subtypes in TCGA cohort. EMT^{high}-AKT subtype had the highest Stromal Score among the four EMT subtypes in 100 iteration repeats; Student's t test was used to analyze statistical significance between 2 groups; ****p < 0.0001; ***p < 0.001; **p < 0.01; *p < 0.05.

B. Comparison of Immune Score among four EMT subtypes in TCGA cohort. EMT^{high}-AKT subtype had the highest Immune score among the four EMT subtypes in 100 iteration repeats; Student's t test was used to analyze statistical significance between 2 groups; ****p < 0.0001; ***p < 0.001; **p < 0.01; *p < 0.05.

C. Comparison of tumor purity among four EMT subtypes in TCGA cohort. EMT^{high}-AKT subtype had the lowest tumor purity among the four EMT subtypes in 100 iteration repeats; Student's t test was used to analyze statistical significance between 2 groups; ****p < 0.0001; ***p < 0.001; **p < 0.01; *p < 0.05.

D. AKT pathway score was negatively correlated with tumor purity in EMT^{high}-AKT subtype in TCGA cohort.

For A-C, left panel: Comparison in pancancer scale; right panel: Summary of 100 iterations, each point represents mean stromal score, immune score or purity in each iteration.

Revised Supplementary Figure 9:

Supplementary Figure 9 Identification of driving pathways

- A. Comparison of FOXO pathway activity among four EMT subtypes in TCGA cohort.
- B. Comparison of HIF pathway activity among four EMT subtypes in TCGA cohort.
- C. Comparison of MAPK pathway activity among four EMT subtypes in TCGA cohort.
- D. Comparison of NOTCH pathway activity among four EMT subtypes in TCGA cohort.
- E. Comparison of TGFB pathway activity among four EMT subtypes in TCGA cohort.
- F. Comparison of TNF pathway activity among four EMT subtypes in TCGA cohort.
- G. Comparison of WNT pathway activity among four EMT subtypes in TCGA cohort.
- H. Comparison of YAP pathway activity among four EMT subtypes in TCGA cohort.
- I. Comparison of AKT pathway activity among four EMT subtypes in TCGA cohort.

For all graphs in this figure, left panel: Comparison in pancancer scale; right panel: Summary of 100 iterations, each point represents mean ssGSEA score in each iteration; Student's t test was used to analyze statistical significance between 2 groups; ****p < 0.0001; ***p < 0.001; **p < 0.01; *p < 0.05; NS p > 0.05.

Revised Supplementary Figure 13:

Supplementary Figure 13 EMT^{high}-AKT subtype tumors exemplify a T cell-dysfunction immune-suppressive phenotype caused by M2-TAMs

A. Comparison of IFNG activity between the four EMT subtypes in TCGA cohort. EMT^{high}-NOS subtype had the highest IFNG activity among the four EMT clusters in 100 iteration times; Student's t test was used to analyze statistical significance between 2 groups; ****p < 0.0001; ***p < 0.001; **p < 0.01; *p < 0.05; NS p > 0.05; left panel: Comparison in pancancer scale; right panel: Summary of 100 iterations, each point represents mean IFNG score in each iteration.

B. Comparison of TNFA activity between the four EMT subtypes in TCGA cohort. EMT^{high}-AKT subtype had the highest TNFA activity among the four EMT clusters in 100 iteration times; Student's t test was used to analyze statistical significance between 2 groups; ****p < 0.0001; ***p < 0.001; **p < 0.01; *p < 0.05; NS p > 0.05; left panel: Comparison in pancancer scale; right panel: Summary of 100 iterations, each point represents mean TNF score in each iteration.

C. GSEA analyses displayed immunosuppressive gene signatures enriched in EMT^{high}-AKT subtype. FDR < 0.05 was considered as statistically significant.

D. Comparison of CD8+ T cell proportion between the four EMT subtypes in TCGA cohort; EMT^{high}-AKT subtype had similar CD8+ T cell infiltration level with other three EMT clusters in 97 iteration times; Student's t test was used to analyze statistical significance between 2 groups; ****p < 0.0001; ***p < 0.001; **p < 0.01; *p < 0.05; NS p > 0.05; left panel: Comparison in pancancer scale; right panel: Summary of 100 iterations, each point represents mean CD8 T cell score in each iteration.

E. Comparison of T-cell dysfunction score between the four EMT subtypes in TCGA cohort; EMT^{high}-AKT subtype had the highest T-cell dysfunction score among the four EMT clusters in 100 iteration times; Student's t test was used to analyze statistical significance between 2 groups; ****p <

0.0001; ***p < 0.001; **p < 0.01; *p < 0.05; NS p > 0.05; left panel: Comparison in pancancer scale; right panel: Summary of 100 iterations, each point represents mean T cell dysfunction score in each iteration.

F. Comparison of M2-TAMs proportion between the four EMT subtypes in TCGA cohort; EMT^{high}-AKT subtype had the highest M2-TAMs fraction among the four EMT clusters in 100 iteration times; Student's t test was used to analyze statistical significance between 2 groups; ****p < 0.0001; ***p < 0.001; **p < 0.01; *p < 0.05; left panel: Comparison in pancancer scale; right panel: Summary of 100 iterations, each point represents mean M2 TAM score in each iteration.

G. Comparison of Tregs proportion between the four EMT subtypes in TCGA cohort. EMT^{high}-AKT subtype had similar Tregs infiltration level with EMT^{high}-AKT subtype in 98 iteration times; Student's t test was used to analyze statistical significance between 2 groups; ****p < 0.0001; ***p < 0.001; **p < 0.01; *p < 0.05; NS p > 0.05; left panel: Comparison in pancancer scale; right panel: Summary of 100 iterations, each point represents mean Tregs score in each iteration.

2. use proper cell line name, eg, "MDA-MB-231" instead of "MDA-231" so as to be consistent with other section of the manuscript. Molecular subtype of breast cancer cell lines should be provided as well (luminal, basal-A, claudin-low).

Response: In this version, we have checked the nomenclature of the cell lines to be consistent with other section of the manuscript and with other work. Besides, we also illustrated the molecular subtypes of breast cancer cell lines in the results part.

Page 15, Line 250:

The human breast cancer cell lines **MDA-MB-231** and **MDA-MB-468** were purchased from the Chinese Academy of Sciences cell bank (Shanghai, China). The human breast cancer cell lines **T-47D**, **HCC38**, human melanoma cell lines **A-375** and **WM-115** were obtained from iCell (Shanghai, China). Human melanoma cell line **MeWo** and **SK-MEL-3** was purchased from FENGHUISHWU (Hunan, China) and COBIER (Jiangsu, China), respectively. The human monocyte cell line **THP-1** was obtained from the Chinese Academy of Sciences cell bank. The mouse breast cancer cell line **4T1** and mouse melanoma cell line **B16-F10** were purchased from Procell Life Science&Technology Co.,Ltd (Wuhan, China). The human breast cancer cell lines **T-47D**, **MDA-MB-468** and human melanoma cell line **A-375** were maintained in Dulbecco's Modified Eagle's Medium (DMEM, Gibco, USA) containing 10% fetal bovine serum (FBS) and 1% penicillin/streptomycin at 37 °C with 5% CO₂. The human breast cancer cell line **MDA-MB-231** was cultured in Leibovitz's L-15 medium containing 10% FBS and 1% penicillin/streptomycin at 37 °C without CO₂. The human breast cancer cell line **HCC38** was cultured in RPMI-1640 medium (Gibco, USA) containing 10% FBS and 1% penicillin/streptomycin (Gibco) at 37 °C with 5% CO₂. The human monocyte cell line **THP-1** was cultured in RPMI-1640 medium (Gibco, USA) containing 10% FBS and 1% penicillin/streptomycin (Gibco) at 37 °C with 5% CO₂. Human melanoma cell line **MeWo** and **WM-115** were cultured in MEM medium (Gibco, USA) containing 10% FBS and 1% penicillin/streptomycin (Gibco) at 37 °C with 5% CO₂. Human melanoma cell line **SK-MEL-3** was cultured in McCoy's 5a medium (Gibco, USA) containing 15% FBS and 1% penicillin/streptomycin (Gibco) at 37 °C with 5% CO₂. The mouse breast cancer cell line **4T1** was cultured in RPMI-1640 medium (Gibco, USA) containing 10% FBS and 1% penicillin/streptomycin (Gibco) at 37 °C with 5% CO₂. Mouse melanoma cell line **B16-F10** was cultured in RPMI-1640 medium (Gibco, USA) containing 10% FBS and 1% penicillin/streptomycin (Gibco) at 37 °C with 5% CO₂.

Page 32, Line 582:

Four breast cancer cell lines (T-47D: EMT^{low} subtype, Luminal A; MDA-MB-468: EMT^{mid} subtype, basal like; HCC38: EMT^{high}-NOS subtype, claudin-low; MDA-MB-231: EMT^{high}-AKT subtype, claudin-low) and four melanoma cancer cell lines (MeWo: EMT^{low} subtype; A-375: EMT^{mid} subtype; SK-MEL-3: EMT^{high}-NOS subtype; WM-115 EMT^{high}-AKT subtype) were used for this purpose.

Revised Figure 4:

Figure 4 AKT pathway is important for mediating mesenchymal transition and immunosuppression only in EMT^{high}-AKT subtype

A. Left panel: Immunoblotting analysis of N-Cadherin and Vimentin indicated that macrophage could not promote mesenchymal transition of T-47D. Right panel: Transwell analysis indicated T-47D could not enhance infiltration ability of macrophages.

B. Left panel: Immunoblotting analysis of N-Cadherin and Vimentin indicated that macrophage could not promote mesenchymal transition of MDA-MB-468. Right panel: Transwell analysis indicated MDA-MB-468 enhanced infiltration ability of macrophages mildly and could not be inhibited by AKT

inhibition.

C. Left panel: Immunoblotting analysis of N-Cadherin and Vimentin indicated that macrophage could partially enhance mesenchymal transition of HCC38 (Vimentin) and could not be inhibited by AKT inhibition. Right panel: Transwell analysis indicated HCC38 enhanced infiltration ability of macrophages mildly and could not be inhibited by AKT inhibition.

D. Left panel: Immunoblotting analysis of N-Cadherin and Vimentin indicated that macrophage could enhance mesenchymal transition of MDA-MB-231 which could be inhibited by AKT inhibition. Right panel: Transwell analysis indicated MDA-MB-231 enhanced infiltration ability of macrophages dramatically and could be inhibited by AKT inhibition.

E. PD-L1 expression did not up-regulate in co-cultured T-47D and macrophages.

F. PD-L1 expression did not up-regulate in co-cultured MDA-MB-468 and macrophages.

G. PD-L1 expression did not up-regulate in co-cultured HCC38 and macrophages.

H. PD-L1 expression up-regulated in co-cultured MDA-MB-231 and macrophages which could be inhibited by AKT inhibition.

For all assays in this figure, tumor cells were pre-treated with DMSO or MK2206 for 48 hours. n = 3; Error bars indicate SD; the Student's t test was used to analyze statistical significance between 2 groups; ns p > 0.05; *p < 0.05; ***p < 0.001; ****p < 0.0001.

Revised Figure 5A:

A

Sample	KNN_cluster	LDA_cluster	LR_cluster
mGSC	Cluster IV	Cluster IV	Cluster IV
4T1	Cluster IV	Cluster IV	Cluster IV
B16-F10	Cluster IV	Cluster IV	Cluster IV

Figure 5A: EMT classification of mGSC, 4T1 and B16-F10 cell line using machine learning method.

Revised Figure 6J-L:

J. MK2206 could overcome anti-PD-L1 resistance in B16-F10 xenograft tumors.

K. MK2206 could overcome anti-PD-1 resistance in B16-F10 xenograft tumors.

L. MK2206 could overcome anti-CTLA-4 resistance in B16-F10 xenograft tumors.

Revised Supplementary Figure 14:

Supplementary Figure 14 AKT pathway is important for mediating mesenchymal transition and immunosuppression only in EMT^{high}-AKT subtype

A. Left panel: Immunoblotting analysis of N-Cadherin and Vimentin indicated that macrophage could not promote mesenchymal transition of MeWo. Right panel: Transwell analysis indicated MeWo could not enhance infiltration ability of macrophages.

B. Left panel: Immunoblotting analysis of N-Cadherin and Vimentin indicated that macrophage could not promote mesenchymal transition of A-375. Right panel: Transwell analysis indicated A-375 enhanced infiltration ability of macrophages mildly and could not be inhibited by AKT inhibition.

C. Left panel: Immunoblotting analysis of N-Cadherin and Vimentin indicated that macrophage could not enhance mesenchymal transition of SK-MEL-3. Right panel: Transwell analysis indicated SK-MEL-3 enhanced infiltration ability of macrophages mildly.

D. Left panel: Immunoblotting analysis of N-Cadherin and Vimentin indicated that macrophage could enhance mesenchymal transition of WM-115 which could be inhibited by AKT inhibition. Right panel: Transwell analysis indicated WM-115 enhanced infiltration ability of macrophages dramatically and could be inhibited by AKT inhibition.

- E. PD-L1 expression did not upregulate in co-cultured MeWo and macrophages.
- F. PD-L1 expression did not upregulate in co-cultured A-375 and macrophages.
- G. PD-L1 expression did not upregulate in co-cultured SK-MEL-3 and macrophages.
- H. PD-L1 expression upregulated in co-cultured WM-115 and macrophages which could be inhibited by AKT inhibition.

For all assays in this figure, tumor cells were pre-treated with DMSO or MK2206 for 48 hours. n = 3; Error bars indicate SD; Student's t test was used to analyze statistical significance between 2 groups; ****p < 0.0001; ***p < 0.001; **p < 0.01; *p < 0.05; NS p > 0.05.

3. How were the cell lines classified into EMThigh/low, AKT etc i didn't see the description or analyses anywhere in the paper.

Response: The detailed method of classifying cell lines into EMT subtypes could be found in the Materials and Methods part.

Page 11, Line 190:

Identification of EMT Subtypes of Non-TCGA Samples

Identification of non-TCGA samples was done with following steps. For human samples: 1) The batch effect between external cohorts and TCGA cohort was removed; 2) Each individual sample in external cohort was scaled with TCGA cohort separately; 3) Euclidian distance between individual samples and four TCGA EMT cluster centers was calculated using the expression of 35 MTCG genes. Euclidian distance= $\sqrt{\varepsilon(nonTCGAi - TCGAi)^2}$, i=35, ε means summation; 4) Individual samples were classified into the corresponding subtype with the shortest distance.

We have also provided a schematic plot of this method in Supplementary Figure 10A.

Supplementary Figure 10A:

Supplementary Figure 10A: Schematic plot to reclassify individual samples into 35 EMTGs-based EMT subtypes.

The EMT results of cancer cell lines could be found in Supplementary Table 15 (CCLE Cluster Result).

Besides, we could also provide the detailed R code of this method to the readers upon request.

Reviewer #2 (Remarks to the Author):

The comments of the reviewers has been adequately addressed and the revised manuscript is significant improved. The current version is acceptable for publication in Communication Biology.

Response: We thank the reviewer for this favorable comment.

REVIEWERS' COMMENTS:

Reviewer #1 (Remarks to the Author):

The manuscript looks good. The authors have addressed my concerns satisfactorily.

Reviewers' comments:

Reviewer #1 (Remarks to the Author):

The manuscript looks good. The authors have addressed my concerns satisfactorily.

Response: We thank the reviewer for this favorable comment.